# Autoantibody Profiling for Accurate Differentiation of Type 1 and Type 2 Diabetes Mellitus in Omani Patients: A Retrospective Study

**DOI:** 10.3390/diagnostics15182296

**Published:** 2025-09-10

**Authors:** Souad Al-Okla, Salima Al Maqbali, Hamdi Al Mutori, Amna Mohammed Al-Hinai, Rayyan Hassan Al Bloushi, Mallak Ahmed Aljabri, Haya Nasser Alsenani, Mohammad Al Shafaee

**Affiliations:** 1College of Medicine and Health Sciences, National University of Science and Technology, P.O. Box 391, Sohar 321, Oman; hamdisaleh@nu.edu.om (H.A.M.); amna200138@nu.edu.om (A.M.A.-H.); rayyan200268@nu.edu.om (R.H.A.B.); mallak200434@nu.edu.om (M.A.A.); haya200369@nu.edu.om (H.N.A.); alshafaee@nu.edu.om (M.A.S.); 2Department of Biology, Faculty of Sciences, Damascus University, Damascus P.O. Box 30621, Syria; 3Department of Pathology & Blood Bank, Suhar Hospital, Ministry of Health, Al Batinah, P.O. Box 49, Sohar 311, Oman; almoq96@yahoo.com

**Keywords:** type 1 diabetes, type 2 diabetes, autoantibodies, anti-GAD, anti-islet, HbA1c, autoimmune diabetes, diagnostic biomarkers, diabetes classification

## Abstract

**Background/Objectives:** Differentiating Type 1 from Type 2 diabetes mellitus (T1DM vs. T2DM) remains clinically challenging, especially in early-onset cases with overlapping features. This study assessed the diagnostic utility of diabetes-related autoantibodies in an Omani cohort and evaluated their predictive performance using machine learning. **Methods:** Clinical and laboratory data from 448 patients (aged ≥ 2 years) in Al Batinah North, Oman, were retrospectively analyzed. We assessed autoantibody positivity (anti-GAD, anti-islet, anti-TPO, anti-tissue), age, sex, and HbA1c. Receiver operating characteristic (ROC) curves and a neural network model were used to evaluate diagnostic accuracy. **Results:** Anti-GAD and anti-islet antibodies were significantly more prevalent in T1DM (69.0% and 64.1%) than T2DM (7.4% and 3.8%; *p* < 0.0001). HbA1c was elevated in both subtypes but lacked discriminatory specificity. Nearly half (48.5%) of T1DM patients showed multiple antibody positivity, especially in younger age groups. Anti-TPO and anti-tissue antibodies were more frequently detected in T1DM, suggesting broader autoimmunity. ROC analysis showed strong predictive value for anti-islet (AUC = 0.835) and anti-GAD (AUC = 0.827). Neural network modeling identified anti-GAD, anti-islet, and age as the most informative predictors, achieving over 92% classification accuracy. Importantly, antibody positivity in a subset of insulin-treated T2DM patients suggested potential latent autoimmune diabetes (LADA) misclassification. **Conclusions:** This is the first study in Oman to combine autoantibody screening with AI-based modeling to refine diabetes classification. Our findings highlight the value of immunological profiling in early diagnosis, uncover possible misclassification, and support AI integration to guide individualized management.

## 1. Introduction

Diabetes mellitus (DM) is a complex and heterogeneous metabolic disorder that represents a public health concern worldwide. In 2021, over 537 million adults were diagnosed with diabetes, with projections indicating an increase to 783 million by 2045 [1]. This rise is particularly severe in the Middle East and North Africa (MENA) region, where rapid urbanization, dietary transitions, and genetic predispositions contribute to some of the highest regional prevalence rates. In Oman, the burden of DM has been increasing steadily, particularly among younger populations and women, with prevalence rates exceeding 20% in some subgroups [2].

Accurate classification of diabetes subtypes, including T1DM, T2DM, and latent autoimmune diabetes in adults (LADA), is essential for ensuring appropriate clinical management and optimizing patient outcomes. T1DM is an autoimmune disease characterized by beta-cell destruction and the presence of specific autoantibodies, such as glutamic acid decarboxylase (GADA), islet cell antibodies (ICA), and insulin autoantibodies (IAA) [3,4,5]. In contrast, T2DM is typically associated with insulin resistance and relative insulin deficiency, without autoimmune involvement [6]. LADA, an adult-onset autoimmune diabetes, often presents with features of T2DM but shares the immunopathological profile of T1DM, leading to frequent misdiagnosis [7,8]. Inaccurate classification can result in inappropriate therapy, poor glycemic control, and increased risk of complications such as retinopathy and cardiovascular disease [9,10,11].

To improve subtype differentiation, autoantibody profiling has been widely utilized in Western populations. The detection of GADA, ICA, and IAA, has been shown to play a critical role in distinguishing between diabetes subtypes and guiding early therapeutic interventions [12,13,14,15]. Additionally, anti-thyroid peroxidase (anti-TPO) and IgA anti-tissue transglutaminase (anti-tTG-IgA) antibodies can detect coexisting autoimmune conditions such as thyroiditis and celiac disease, commonly associated with T1DM [16,17]. Large cohort studies including TEDDY have confirmed the predictive value of autoantibodies for disease onset and subtype identification [18], However, Oman lacks localized data to guide clinical decision-making. Factors such as high consanguinity rates and overlapping clinical features between diabetes subtypes further complicate accurate diagnosis and underscore the need for population-specific evidence [2,19]. Therefore, this study directly addresses these gaps by retrospectively analyzing four key autoantibodies (anti-GAD, anti-islet, anti-TPO, and anti-tTG IgA) in a cohort of Omani patients aged ≥ 2 years from Al Batinah North. Our approach includes thorough stratification by age and sex and leverages explainable AI techniques to develop a robust, locally validated classification tool for diabetes subtypes. This represents a pioneering effort in Oman, introducing an innovative, data-driven framework that enhances diagnostic precision, supports personalized treatment, and informs national diabetes management policies.

## 2. Materials and Methods

### 2.1. Study Design and Population

This cross-sectional study was conducted using retrospective data extracted from the Al Shifa electronic medical record system at Sohar Hospital, Oman, spanning the period from 2020 to 2024.

Sohar Hospital functions as a tertiary referral center for diabetes care in the region. In this setting, autoantibody testing is not part of routine screening but is typically ordered selectively by specialists, particularly in pediatric patients or when the clinical presentation suggests autoimmune diabetes. In routine clinical practice, most patients with T2DM are managed at the primary care level, with referral to Sohar Hospital generally reserved for complex or atypical cases requiring further diagnostic evaluation, optimization of management, or initiation of advanced pharmacotherapy. By contrast, most patients with T1DM present with diabetic ketoacidosis (DKA), necessitating hospitalization and comprehensive investigation. This referral and testing pattern influenced the composition of the study cohort, resulting in a higher proportion of T1DM patients, while many T2DM cases, typically managed in primary care, may not be captured unless referred for further evaluation.

The study included patients of both sexes and all age groups with a confirmed diagnosis of diabetes mellitus and available autoantibody test results. A total of 448 eligible patients with complete clinical and laboratory data were included in the final analysis.

### 2.2. Eligibility Criteria

Inclusion criteria were as follows: (1) a confirmed diagnosis of diabetes mellitus between 2020 and 2024; (2) age ≥ 2 years; (3) availability of results for at least one autoantibody screening test, including anti-GADA, anti-islet (islet cell antibody; anti-ICA), anti-tissue transglutaminase, or anti-TPO antibodies; (4) Omani nationality to ensure genetic and cultural homogeneity.

Exclusion criteria included: (1) gestational diabetes; (2) neonatal or monogenic diabetes; (3) post-transplant diabetes; (4) individuals without a diabetes diagnosis; (5) patients with autoimmune diseases unrelated to diabetes or the antibodies under investigation.

### 2.3. Classification and Stratification

Patients were classified as T1DM or T2DM based on the clinical diagnosis recorded in the electronic medical record (EMR), as assigned by the treating physician during the initial consultation and confirmed in follow-up visits. This classification was made independently of autoantibody results and reflected routine clinical criteria. In our setting, T1DM was typically diagnosed in patients presenting with DKA or ketosis at onset, younger age at presentation, lower or normal BMI, and early requirement for insulin. T2DM was generally diagnosed in adults with gradual symptom onset, overweight/obesity, and initial response to oral glucose-lowering agents. Autoantibody profiles were analyzed as independent variables to assess their diagnostic value and were not used to define the original classification.

Additional subgroups included the following: unspecified diabetes (diagnosed diabetes without sufficient clinical or serological data for precise classification), isolated hyperglycemia (elevated glucose without a formal diabetes diagnosis), and normal glycemia (normal glucose and HbA1c levels without a diabetes diagnosis, often present due to screening or referral). These three groups were excluded from subtype comparisons due to the absence of antibody testing. Included patients were further stratified into four age groups: childhood (<13 years), adolescence (13–19 years), adulthood (20–35 years), and elderly (>35 years).

### 2.4. Data Collection

Clinical and laboratory data were retrospectively retrieved from patient records. Demographic variables included age at the time of autoantibody testing and sex. Clinical variables included diabetes subtype, treatment regimen (e.g., insulin, sulphonylureas, metformin, sitagliptin), and diabetes-related complications such as DKA, chronic kidney disease (CKD), retinopathy, neuropathy, and diabetic foot. Because data were extracted from electronic records of hospital visits during 2020–2024, chronic complications (retinopathy, neuropathy, chronic kidney disease, diabetic foot) were captured as documented diagnoses within this period; a uniform screening protocol was not undertaken. DKA was recorded when present at the time of hospital presentation within the study window. In most cases, autoantibody testing was performed shortly after diagnosis and, for suspected T1DM cases, before insulin initiation to avoid interference with the anti-islet assay.

Laboratory data included HbA1c and autoantibody test results for anti-GAD, anti-islet, anti-tissue, and anti-TPO antibodies. HbA1c values were classified using a threshold of ≥8.0% (64 mmol/mol), commonly associated with poor glycemic control. Autoantibody results were categorized as positive or negative based on established diagnostic thresholds:Anti-GAD: ≥5 U/mLAnti-islet: ≥1 U/mLAnti-tissue: ≥20 U/mLAnti-TPO: ≥60 U/mL

The number of positive autoantibodies per patient was also recorded to assess cumulative autoantibody burden.

### 2.5. Statistical Analysis

Descriptive statistics were employed to summarize the demographic, clinical, and laboratory characteristics of the study population. Normality of continuous variables (e.g., autoantibody levels and HbA1c) was evaluated using the Kolmogorov–Smirnov (K–S) test, and these results informed the choice of parametric or nonparametric comparisons. Categorical variables, such as autoantibody positivity rates, were compared between groups using the Chi-squared test. For continuous variables that did not follow a normal distribution (e.g., autoantibody signal intensities), the Mann–Whitney U test was used to assess group differences. Relationships between continuous variables, such as age and autoantibody levels, were examined using Spearman’s rank correlation coefficient (r).

To evaluate the diagnostic performance of each biomarker in differentiating diabetes subtypes, ROC curve analysis was conducted, with calculation of the area under the curve (AUC), sensitivity, and specificity. Additionally, a TURF (Total Unduplicated Reach and Frequency) analysis was performed to identify the most efficient combinations of autoantibodies for broad detection coverage. HbA1c was also included in the TURF, ROC, and neural network analyses to allow comparison of its diagnostic contribution relative to immunological markers. Its inclusion was exploratory, acknowledging that HbA1c is not a subtype-specific biomarker but may provide contextual information on glycemic burden at the time of testing. This exploratory tool helps evaluate the incremental diagnostic value of each marker and their combinations.

A multilayer perceptron artificial neural network (ANN) model was also developed to predict diabetes subtype using key variables: age group, sex, HbA1c, and autoantibody test results. These methods offer a data-driven approach to detect patterns that may not be captured by traditional hypothesis testing, particularly in complex or overlapping phenotypes such as LADA. Prior to modeling, all predictors were standardized; scaling parameters were estimated from the training set and applied to the test set. The study cohort (*N* = 448) was randomly partitioned at the patient level using stratified sampling by diabetes subtype into non-overlapping training (70%; *n* = 314) and testing (30%; *n* = 134) sets. Model selection and tuning were performed within the training set without access to test labels, and the held-out test set was evaluated once for final performance. Performance was summarized using prediction error rate, cross-entropy loss, and AUC, and the relative contribution of each predictor to classification accuracy was computed.

The ANN was trained and validated using the Multilayer Perceptron module in SPSS Statistics version 26 (IBM Corp., Armonk, NY, USA). The network architecture included an input layer with one neuron per predictor variable, a single hidden layer with an automatically determined number of neurons, and an output layer using the softmax activation function for multi-class classification. The model was trained using a scaled conjugate gradient algorithm with a maximum of 50 epochs. Internal validation was achieved through continuous monitoring of cross-entropy loss, and early stopping was applied to prevent overfitting.

All statistical analyses were conducted using SPSS Statistics version 26, while all visualizations were created using GraphPad Prism^®^ version 10 (GraphPad Software, San Diego, CA, USA). A two-tailed *p*-value < 0.05 was considered statistically significant throughout the analysis.

## 3. Results

### 3.1. Distribution of Diabetes Subtypes in the Cohort

Among the 448 patients included in the analysis, 299 (66.7%) were classified as having T1DM, and 84 (18.8%) as T2DM. The remaining 65 patients (14.5%) exhibited features that did not clearly align with either subtype—such as indeterminate autoantibody profiles, mixed treatment regimens, or possible latent autoimmune diabetes in adults (LADA) without definitive criteria—and were therefore excluded from subtype-specific comparisons. In the T1DM group, 52.2% were male (156/299) and 47.8% female (143/299); the T2DM group showed a similar distribution, with 51.2% males (43/84) and 48.8% females (41/84). These findings highlight the predominance of T1DM in this cohort, consistent with the younger age profile of the studied population.

### 3.2. Association Between Age and Type of Diabetes Diagnosis

The ages of patients ranged from 2 to 83 years (mean ± SD: 23.7 ± 15.4), with a median age of 19 years. As shown in Figure 1, patients diagnosed with T1DM were significantly younger than those with T2DM. The mean age was 19.3 ± 12.7 years for T1DM and 32.9 ± 16.2 years for T2DM, with corresponding median ages of 15 and 37 years, respectively (*p* < 0.0001).

By age band, the cohort comprised <13 years (*n* = 133), 13–19 years (*n* = 97), 20–35 years (*n* = 109), and >35 years (*n* = 109). T1DM was most prevalent in the youngest group (<13 years: 96.4%) and decreased progressively with age, whereas the proportion of T2DM increased with age, peaking in the >35 years band (53.6%). Subtype distribution differed markedly across age bands (overall *p* < 0.0001), with no significant between-subtype difference in 13–19 (*p* = 0.08) and 20–35 (*p* = 0.131).

### 3.3. Diagnostic Performance of Diabetes Biomarkers

This study analyzed data of 448 patients with available data on key autoantibodies and HbA1c levels. The mean antibody levels (±standard deviation) were as follows: anti-GAD: 403.7 ± 705.8 U/mL (*n* = 442), anti-islet: 7.5 ± 8.6 U/mL (*n* = 430), anti-tissue: 12.6 ± 38.4 U/mL (*n* = 210), anti-TPO: 210.5 ± 408.9 U/mL (*n* = 147), and HbA1c: 9.95 ± 2.6% (n = 444). Based on Kolmogorov–Smirnov tests, all biomarkers except HbA1c (*p* = 0.09) deviated significantly from a normal distribution.

HbA1c, measured using a quantitative assay, demonstrated acceptable reproducibility with a coefficient of variation (CV) of 19.36%. The diabetes-related autoantibodies (anti-GAD, anti-islet, anti-TPO, and anti-tissue) were measured using semi-quantitative ELISA kits, which report results as optical density-based relative units rather than absolute concentrations; therefore, CV values are reported only for HbA1c Positivity for each biomarker is reported individually in Table 1 according to the manufacturer-specified thresholds.

### 3.4. HbA1c Levels and Positivity by Sex and Age Groups

Figure 2 shows HbA1c levels and positivity rates across sex and age groups in T1DM and T2DM patients. In the entire cohort (Total), T1DM patients had significantly higher mean HbA1c levels than T2DM (*p* < 0.001), a pattern consistent across females (*p* < 0.001) and males (*p* < 0.01). Mean HbA1c values in T1DM remained consistently elevated across all age groups, ranging from 10.0% to 10.9%, with positivity rates above 85%, peaking at 90% in patients <13 years. Notably, this youngest group showed significantly higher positivity than the 13–19 age group in both the total cohort and T1DM subgroup (*p* < 0.001).

In T2DM, mean HbA1c ranged from 8.3% to 9.7%, with positivity increasing with age, from 38.5% in 13–19 years to 69.6% in >35 years (*p* = 0.05). No significant HbA1c differences were found between males and females within either subtype.

These findings highlight the consistently elevated glycemic burden in T1DM, particularly among younger patients, and reinforce the need for age-stratified evaluation in diabetes management.

### 3.5. Autoantibody Distribution by Diabetes Subtype

As shown in Figure 3, strong and statistically significant differences in autoantibody profiles were observed between T1DM and T2DM patients. Anti-GAD and anti-islet antibodies demonstrated significantly higher positivity rates in the T1DM group (69.0% and 64.1%, respectively) compared to T2DM (7.4% and 3.8%; *p* < 0.0001). In addition to frequency, the levels of these antibodies were also significantly elevated in T1DM, underscoring their strong association with autoimmune pathophysiology.

In contrast, anti-tissue and anti-TPO antibodies showed low overall prevalence in both subtypes, with positivity rates of 9.6% and 28.7% in T1DM, respectively, and were nearly absent in T2DM (0% and 1.2%, %, respectively). This suggests that while these antibodies may reflect broader autoimmune activation, they have limited value for subtype differentiation.

Violin plots illustrate the distribution of measured antibody levels, with widths indicating data density; inside each violin, the dashed vertical line marks the group median (50th percentile).

### 3.6. Anti-GAD and Anti-iIslet Diagnostic Profiles Across Age and Sex

As illustrated in Figure 4A,B, anti-GAD and anti-islet antibody levels were significantly higher in T1DM compared with T2DM across all age groups and sexes (*p* < 0.0001). Analyses are shown for All (entire tested cohort) and Total (confirmed T1DM and T2DM), stratified by sex and age group (<13, 13–19, 20–35, >35 years). For anti-GAD, patients with T1DM exhibited high mean levels (typically >500 U/mL) across age groups and both sexes, whereas T2DM values were generally <50 U/mL; no sex-based differences were detected within the subtype. Anti-GAD positivity (≥5 U/mL) was 69.0% in T1DM and 7.2% in T2DM; within T1DM, positivity declined with age (<13 compared with >35, *p* < 0.05; 13–19 compared with >35, *p* < 0.05) (Figure 4A).

Anti-islet showed a similar pattern. In the Total cohort, positivity was 64.1% in T1DM compared with 3.8% in T2DM (*p* < 0.0001); within T1DM, both levels and positivity were highest in <13 and decreased with age (<13 compared with 13–19, *p* = 0.012; <13 compared with 20–35 and >35, both *p* < 0.001), while T2DM remained low across age and sex (Figure 4B). These analyses describe group-level distributions rather than predictive modeling.

### 3.7. Antibody Profiles Indicating Comorbid Autoimmunity in T1DM and T2DM

In contrast to the diagnostic markers (anti-GAD/anti-islet), anti-tissue and anti-TPO were evaluated to characterize comorbid autoimmunity associated with T1DM Analyses are shown for All (entire tested cohort) and Total (confirmed T1DM and T2DM), stratified by sex and age group (<13, 13–19, 20–35, >35 years)**,** as shown in Figure 5A,B. For anti-tissue, levels were generally low in Total, with 91.9% of patients below the ≥20 U/mL threshold; mean levels were higher in T1DM (14.4 ± 41.5 U/mL) than in T2DM (3.1 ± 3.7 U/mL), but this difference was not statistically significant. Within T1DM, females showed slightly higher levels and positivity than males (12% and 6.5%), without a statistically significant difference. Within T1DM, females showed slightly higher levels and positivity (12% and 6.5%), although no sex-based differences reached significance. All positives occurred in T1DM and were confined to patients <35 years, particularly 20–35; no positivity was observed in >35 or in T2DM (Figure 5A). For anti-TPO, overall positivity was 27.9%; positivity was more frequent in T1DM (28.7%) than T2DM (14.3%) without a significant difference. Within T1DM, male and female positivity rates were similar; the highest positivity appeared in <13 and 13–19 years, but no clear age-dependent pattern was observed. Mean anti-TPO levels peaked in 20–35 years without a corresponding rise in positivity, and T2DM showed minimal expression across age groups (Figure 5B).

### 3.8. Patterns of Autoantibody Associations in Relation to Age and Glycemic Control

Figure 6 illustrates the correlations among diabetes-related autoantibodies and their associations with age and metabolic markers. A strong and statistically significant correlation was observed between anti-islet and anti-GAD antibodies (r = 0.63, *p* < 0.0001), indicating frequent co-expression of islet-specific autoantibodies and supporting their central role in pancreatic autoimmunity, particularly in T1DM.

As expected, thyroid (anti-TPO) and systemic (anti-tissue) autoantibodies showed no significant correlation with islet-specific markers, consistent with their known role as comorbid, organ-specific antibodies rather than direct diagnostic markers for T1DM. A weaker but significant correlation was found between anti-tissue and anti-GAD antibodies (r = 0.14, *p* < 0.05), suggesting limited overlap between systemic and β-cell–specific immune responses. In contrast, no significant correlations were observed between anti-TPO and either anti-GAD or anti-islet antibodies, nor between anti-tissue and anti-islet or anti-TPO antibodies.

HbA1c levels showed modest but significant positive correlations with anti-islet (r = 0.15, *p* < 0.005) and anti-GAD (r = 0.14, *p* < 0.005) antibodies, suggesting a potential association between glycemic control and islet autoimmunity. No such correlations were found with anti-TPO or anti-tissue antibodies.

Age-stratified analysis revealed distinct patterns in autoantibody expression. Age was inversely correlated with anti-GAD and anti-islet (r = −0.31, *p* < 0.001 for both) and positively correlated with anti-TPO (r = 0.36, *p* < 0.001), suggesting that islet-specific autoimmunity predominates at younger ages, while thyroid autoimmunity becomes more prevalent with aging. No significant correlations were found between age and either anti-tissue antibody levels or HbA1c.

### 3.9. Patterns and Burden of Autoantibody Positivity by Diabetes Subtype

The distribution of autoantibody positivity counts revealed significant differences between diabetes subtypes (Figure 7). T1DM patients exhibited a significantly higher frequency of multiple autoantibody positivity (*p* < 0.001). Dual positivity was the most common pattern (48.5%), followed by single (18%), triple (8.5%), and quadruple (1.4%) positivity. Only 23.7% of T1DM patients were negative for all tested autoantibodies.

In contrast, 90.5% of T2DM patients were negative for all autoantibodies, significantly more than in T1DM (*p* < 0.001). Single and dual positivity were observed in only 7.1% and 2.4% of T2DM patients, respectively (*p* < 0.05 and *p* < 0.001 compared to T1DM), while triple and quadruple positivity were absent. These findings underscore the higher autoantibody burden and more complex immunological profile characteristic associated with T1DM.

Additional insight into age- and sex-specific patterns of autoantibody burden is provided in Appendix A, which confirms that younger T1DM patients and both sexes exhibit significantly higher antibody counts, consistent with trends described in the main analysis.

### 3.10. Autoantibody Burden Across Treatment Modalities and Complications

Autoantibody burden, measured as the average number of positive tests per patient, varied significantly by both treatment type and the presence of diabetes-related complications (Table 2; Appendix A).

Patients receiving insulin therapy exhibited the highest autoantibody burden (mean 1.56 ± 0.05), with T1DM patients showing significantly greater mean positivity (1.74 ± 0.05) compared to T2DM (0.53 ± 0.08; *p* < 0.0001). Insulin-treated individuals accounted for nearly 90% of all positive cases. In contrast, patients on oral therapies showed markedly lower levels of autoantibody positivity, with mean values of 0.37 ± 0.13 for sulfonylureas (1.8% of the cohort), 0.30 ± 0.05 for metformin (23.2%), and 0.21 ± 0.05 for sitagliptin (13.5%). No significant differences between T1DM and T2DM were observed within these oral treatment groups, where positivity consistently remained below 0.6.

When analyzed by complications, DKA emerged as the most frequent (32.5% of patients; 89.4% T1DM) and was associated with the highest autoantibody burden (1.85 ± 0.10), significantly greater in T1DM (2.01 ± 0.12) than in T2DM (1.00 ± 0.11; *p* < 0.0001). In contrast, other complications showed considerably lower autoimmune reactivity: CKD (3.4%) with a mean of 0.34 ± 0.06, retinopathy (4.7%) 0.50 ± 0.07, neuropathy (2.4%) 0.29 ± 0.06, and diabetic foot (1.3%) the lowest at 0.15 ± 0.05. All were significantly lower than DKA (*p* < 0.0001) and below the cohort average (*p* < 0.01), with no differences between T1DM and T2DM.

These findings underscore that autoimmune reactivity is most pronounced in insulin-treated patients and in those experiencing acute metabolic decompensation, particularly DKA, while other complications show limited association with antibody burden.

### 3.11. Integrated Autoantibody and Predictive Modeling Analysis

To comprehensively evaluate the diagnostic utility of laboratory and immunological markers for diabetes classification, we employed both traditional and AI-driven analyses (Figure 8). A TURF analysis identified the combination of anti-GAD and anti-islet antibodies as the most effective, covering 92.4% of all antibody-positive cases and 85.1% among individuals with complete profiles. Anti-GAD alone was the most prevalent marker (48.2% positivity) and appeared in nearly all high-yield combinations, emphasizing its pivotal role in autoimmune diabetes detection. Dual-marker combinations involving anti-GAD and either anti-TPO or anti-tissue antibodies also demonstrated broad coverage (>70%). However, combinations that did not include anti-GAD, such as anti-TPO with anti-tissue, captured less than 50% of cases, highlighting the limited incremental benefit of including these markers independently. These findings support the prioritization of anti-GAD, especially in combination with anti-islet antibodies, for efficient autoimmune diabetes screening (Figure 8A).

To further assess individual marker performance, ROC curve and AUC analyses were conducted for five biomarkers (Figure 8B). Anti-GAD, anti-islet, and HbA1c exhibited the highest diagnostic accuracy for T1DM, with AUCs of 0.91, 0.83, and 0.84, respectively (all *p* < 0.0001). These markers had limited discriminative ability for T2DM (AUCs < 0.5). Anti-TPO and anti-tissue antibodies showed high specificity but low sensitivity, reducing their standalone diagnostic value. In the sensitivity–specificity bubble plot (Figure 8C), only anti-GAD and anti-islet clustered within the optimal quadrant, indicating a favorable diagnostic balance.

Building on these results, we developed a neural network model incorporating demographic and immunological variables (Figure 8D). The model demonstrated high accuracy with mean AUC of 0.927 ± 0.03 and low prediction errors (training: 4.1%, testing: 4.6%). Anti-GAD was the most significant predictor (*p* < 0.001), followed by anti-islet and age (*p* < 0.05 to *p* < 0.001), whereas HbA1c, anti-TPO, and sex contributed minimally. These findings affirm the value of islet-related antibodies and highlight the added diagnostic precision afforded by machine learning.

## 4. Discussion

Accurate subclassification of diabetes remains a global diagnostic challenge, particularly in adults, where overlapping clinical features between T1DM and T2DM can obscure accurate diagnosis. This is especially relevant in regions like Oman, where comprehensive immunological profiling is not yet standard practice. To our knowledge, this study is among the first in the region to combine broad-spectrum autoantibody testing with advanced computational modeling, offering novel insights into the diagnostic value of autoimmune markers within an Omani population. Given the vastly different treatment strategies and prognoses between T1DM and T2DM, accurate classification is not only essential for appropriate clinical management but also pivotal for reducing long-term complications and optimizing outcomes.

A key contextual consideration is the imbalance in the number of T1DM and T2DM cases included in the cohort. The predominance of T1DM in our sample likely reflects the nature of the study setting, a tertiary referral center, where autoantibody testing is selectively requested for patients with suspected autoimmune diabetes, particularly children and those with atypical presentations. In contrast, most T2DM patients in Oman are managed at the primary care level and are not typically referred for autoantibody testing unless they exhibit poor response to therapy. This referral pattern may contribute to underrepresentation of T2DM cases and should be considered when interpreting model generalizability and antibody prevalence findings.

We found that anti-GAD and anti-islet antibodies were significantly more prevalent in T1DM (69% and 64.1%) than in T2DM (7.4% and 3.8%), with no significant differences observed between male and female patients. These findings are consistent with international studies reporting similar autoantibody profiles in T1DM populations. For example, Christie et al. [20] reported anti-islet antibody positivity rates of 73–75% in Swedish children, while Pardini et al. [21] found anti-GAD and anti-IA-2 positivity in 80% and 62.9% of Brazilian T1DM patients, respectively. Similarly, Hazime et al. [22], noted that approximately 75% of T1DM patients in Morocco tested positive for at least one diabetes-related autoantibody, with anti-GAD being the most common. The concordance between our findings and international data reinforces the diagnostic utility of anti-GAD and anti-islet antibodies across diverse populations.

Importantly, the presence of anti-GAD and anti-islet antibodies in a small subset of patients clinically classified as T2DM (7.4% and 3.8%) raises concern for possible diagnostic misclassification. These individuals may represent LADA, the form of autoimmune diabetes that mimics T2DM in its early stages due to its adult onset and slow progression. Delayed recognition of LADA can lead to postponed insulin initiation, suboptimal glycemic control, and elevated risk for complications. Therefore, integrating routine antibody screening into clinical workflows, particularly for adult-onset diabetes, could substantially enhance diagnostic precision and treatment planning.

Our findings also align with previous clustering studies indicating that early seroconversion to insulin autoantibodies (IAA) and GAD is predictive of rapid disease progression [23]. The high reproducibility and robust diagnostic performance of anti-GAD and anti-islet antibodies observed in this study support their application in routine screening protocols, particularly in settings like Oman where LADA may be under-recognized [24,25,26].

In our cohort, T1DM patients under 13 years of age exhibited the highest autoantibody burden, with a mean of 1.69 to 1.76 positive autoantibodies per patient, reflecting intense autoimmune activity in early-onset disease. Additionally, we observed a clear inverse correlation between age and antibody positivity, as the prevalence of anti-GAD and anti-islet antibodies declined with increasing age. These findings are consistent with previous research showing a higher autoantibody load in pediatric populations [27]. Similar patterns were reported by the TrialNet and BABYDIAB cohorts, which found clustering of multiple autoantibodies in young children at risk for or newly diagnosed with T1DM [23].

The strong diagnostic value of anti-islet antibodies in identifying T1DM, especially in pediatric patients, is supported by previous research [28,29]. Similarly, anti-GAD positivity is notably higher in younger individuals with T1DM, reaching up to 80% in children according to Pardini et al. [21] and Sabbah et al. [30]. GAD titers tend to peak during the early stages of autoimmune diabetes and decline with age, further supporting their use as early markers of disease activity [31].

Conversely, anti-GAD positivity in T2DM was minimal across all age groups, consistent with previous findings by Schiel & Müller [32] and Martinka et al. [33], who reported rare cases of positivity suggestive of possible LADA. In our study, 90.5% of T2DM patients were negative for all tested autoantibodies, emphasizing the need for systematic immunological screening to prevent subtype misclassification and ensure appropriate treatment.

HbA1c levels were significantly higher in T1DM patients across all demographic groups, with the most pronounced elevations in children under 13 years, over 80% of whom had values exceeding 10.0% (86 mmol/mol). This finding aligns with prior studies linking elevated HbA1c with greater glycemic variability and early β-cell dysfunction in T1DM. Studies have also associated high HbA1c levels with poorer metabolic control and more rapid disease progression in pediatric populations [34,35]. While HbA1c is not subtype-specific, it offers valuable insight into glycemic burden and disease severity at the time of testing. In our study, its inclusion enabled the comparison of metabolic status across diabetes subtypes and age groups. Its limited discriminatory power in ROC and ANN models reinforces the need for immunological profiling to support accurate classification. This limitation highlights the need to combine metabolic assessment with immunological profiling for more accurate subtype classification.

In this context, our findings on the distribution of diabetes-related autoantibodies offer further diagnostic clarity. Although anti-TPO and anti-tissue antibodies were less frequently observed overall, they were more commonly detected in T1DM patients, especially in younger females. Anti-TPO positivity reached 27.9%, peaking in the 20–35 age group, though no statistically significant differences were found across sex or age categories. This pattern aligns with earlier studies documenting elevated autoimmune thyroid markers in diabetic populations [36,37]. However, as emphasized by Prázný et al. [38] and Shliakhova & Chumak [39], the diagnostic utility of anti-TPO in diabetes classification is limited, serving better as an indicator of comorbid thyroid autoimmunity rather than a differentiator between T1DM and T2DM.

Similarly, although anti-tissue antibodies had low overall positivity (<10%), they showed modest elevations in females and younger T1DM patients, consistent with findings from Williams et al. [40] and Al-Hakami [41]. These results suggest that anti-tissue antibody testing may have greater relevance for screening autoimmune comorbidities such as celiac disease, rather than for distinguishing diabetes subtypes.

To better understand the immunological landscape, we explored the relationships among various autoantibodies. A strong correlation was observed between anti-GAD and anti-islet antibodies (r = 0.63, *p* < 0.0001), with a weaker but significant association between anti-GAD and anti-tissue antibodies (r = 0.14, *p* < 0.05). These correlations suggest overlapping autoimmune mechanisms, consistent with prior research demonstrating co-expression of multiple autoantibodies in individuals predisposed to autoimmune diabetes [42,43].

This observation was supported by positivity patterns within our cohort. Nearly half (48.5%) of T1DM patients were dual-positive for autoantibodies, and an additional 10% had three or more positive markers. These multi-autoantibody profiles were exclusive to the T1DM group, highlighting their value for predicting progression and staging disease severity [27,44]. Their presence not only reflects more complex autoimmunity but also provides a critical tool for risk stratification, particularly when integrated into predictive models [42].

To optimize the use of these biomarkers in clinical practice, we examined their combined diagnostic utility. TURF analysis revealed that anti-GAD and anti-islet antibodies together captured 92.4% of all antibody-positive cases, making them the most efficient pair for screening. This aligns with prior studies by Verge et al. [45] and Wasserfall & Atkinson [46], who emphasized their diagnostic strength. ROC analysis further confirmed high discriminative value for T1DM (AUCs > 0.82), but poor performance for T2DM, reaffirming the specificity of these markers for autoimmune diabetes. HbA1c was also included in the TURF analysis on an exploratory basis to compare its contribution with that of immunological markers; as expected, its predictive value for subtype discrimination was minimal, consistent with its known limitations as a diagnostic marker.

Collectively, these findings strongly support the implementation of comprehensive autoantibody panels in routine clinical workflows. Such panels can improve diagnostic precision, reduce misclassification of LADA, and enable more timely and individualized treatment strategies.

Building upon these diagnostic insights, our treatment-based analysis further revealed strong immunological evidence suggestive of diabetes misclassification, particularly among insulin-treated patients initially labeled as T2DM. Autoantibody burden in this group was significantly elevated, with insulin-treated individuals exhibiting a mean of 1.56 positive markers. T1DM patients had the highest burden (mean 1.74), as expected. However, insulin-treated T2DM patients showed a mean positivity of 0.53, which was significantly higher than those managed exclusively with oral agents (all <0.4). This unexpected autoantibody presence is inconsistent with classic T2DM and supports a misdiagnosed latent autoimmune diabetes in adults (LADA) phenotype. These findings align with observations from Hathout et al. (2001) [10], who showed that autoimmune markers are rarely present in youth clinically diagnosed with T2DM unless misclassification or latent autoimmune diabetes is involved. Similarly, Ravikumar and Mishra et al. [7,8] emphasized the diagnostic value of autoantibodies in distinguishing LADA from classic T2DM, particularly in insulin-requiring cases. Our findings reinforce this by showing that most antibody-positive cases were concentrated in the insulin-treated group, highlighting the diagnostic value of autoantibody screening in guiding treatment. Misclassified patients may be inappropriately managed with oral agents despite underlying β-cell autoimmunity, delaying effective intervention.

To better understand the clinical implications of this elevated autoimmune burden, we next explored its relationship with diabetes-related complications. DKA, the most frequent acute complication in our cohort (32.5%), was associated with the highest mean number of positive antibodies (1.85), with T1DM patients averaging 2.01, significantly higher than those with T2DM (1.00; *p* < 0.0001). This strong association reinforces the concept that elevated autoimmune activity contributes to metabolic instability, consistent with findings by Krzewska & Ben-Skowronek [16] and Popoviciu et al. [17], who linked autoimmune burden to increased risk of DKA. Similarly, Derrou et al. [44] demonstrated that multiple autoantibody positivity is associated with increased complication rates in T1DM.

Together, the analyses of treatment patterns and complication profiles provide converging evidence of misclassification in autoimmune diabetes. They also highlight the practical value of autoantibody screening not only for accurate diagnosis but also for risk stratification, particularly in populations where T1DM and LADA are often underrecognized or misdiagnosed as T2DM.

To enhance diagnostic precision, we developed a multilayer perceptron neural network that integrated age, sex, HbA1c, and autoantibody profiles. The model demonstrated robust diagnostic performance (AUC = 0.927), with anti-GAD, anti-islet antibodies, and age emerging as the most influential variables. Anti-GAD alone showed significantly greater predictive importance than all other features (*p* < 0.001), while HbA1c, sex, and anti-tissue antibodies contributed only marginally to subtype discrimination. These findings emphasize that while metabolic markers like HbA1c are valuable for disease monitoring, they are insufficient for accurate subtype classification without immunological context. The model outperformed conventional classification approaches, underscoring the utility of machine learning for supporting clinical decision-making in complex diagnostic settings [47,48]. Although these statistical approaches, such as TURF and neural network modeling, are less commonly applied in clinical diabetes research, they provide a valuable framework for integrating immunological and demographic data. This strategy helps bridge the gap between laboratory data and clinical diagnosis, particularly in settings with diagnostic uncertainty.

Importantly, this study represents one of the first applications of artificial intelligence in immunologically informed diabetes classification in the Middle East. By combining computational modeling with targeted biomarker screening, this strategy offers a practical, scalable approach for improving diagnostic workflows, particularly in settings where misclassification of T1DM and LADA is common and access to specialist diagnostic services may be limited.

The artificial neural network model developed in this study demonstrated high predictive performance for diabetes subtype classification, achieving an average AUC of 0.927 ± 0.03 and a low testing error rate of 4.6% in internal validation. These findings underscore the potential of AI-driven modeling, particularly when incorporating islet-specific autoantibodies, to enhance diagnostic precision in complex phenotypes such as LADA.

Despite the strengths of this study, several limitations should be acknowledged:

First, the absence of IA-2 and ZnT8 autoantibodies represents a key limitation. These markers are known to improve the sensitivity of autoimmune diabetes detection and are particularly useful for identifying patients with multiple autoantibody positivity. However, at Sohar Hospital, these tests were not part of the routine diagnostic protocol during the study period and were not requested by the adult or pediatric departments. As such, they were unavailable for inclusion in this retrospective analysis. Future prospective studies incorporating a broader panel of autoantibodies may yield a more comprehensive immunological profile and enhance diagnostic accuracy.

Second, C-peptide measurements, which are critical for assessing residual β-cell function, were sparsely and selectively available. This precluded their use in stratified analyses or integration into machine learning models. The limited availability reflects current clinical practice in our setting, where C-peptide testing is reserved for specific diagnostic dilemmas. Future research should incorporate standardized C-peptide assessment at or near diagnosis to better evaluate β-cell reserve and distinguish between diabetes subtypes.

Third, the study did not stratify results based on diabetes duration due to incomplete documentation of disease onset in the electronic medical records, especially among patients referred from external centers. Disease duration is a critical factor in interpreting antibody persistence and differentiating latent autoimmune diabetes in adults (LADA) from long-standing T2DM. Prospective studies with accurate and standardized documentation of disease onset are needed to address this gap.

Fourth, the artificial neural network model developed in this study, while showing high predictive performance (AUC = 0.927 ± 0.03; testing error rate = 4.6%), was validated only on an internal dataset. It was not tested on an external or independent cohort. As such, the generalizability of this model to other populations and healthcare settings remains uncertain. External validation using larger, multicenter datasets with diverse demographic and clinical characteristics is essential before clinical application.

Finally, it is important to note that only islet-specific autoantibodies (anti-GAD and anti-islet) were used to inform diabetes classification in this study. Other antibodies, such as anti-TPO and anti-tissue transglutaminase, were included solely to explore the presence of comorbid autoimmune conditions associated with T1DM, not for diagnostic differentiation. Their detection was interpreted as evidence of broader autoimmune activity rather than being directly linked to diabetes subtype classification.

## 5. Conclusions

This study highlights the diagnostic power of autoantibody profiling, particularly anti-GAD and anti-islet antibodies, for accurately distinguishing of diabetes subtypes in the Omani population. The high prevalence and multi-autoantibody patterns in T1DM, particularly among younger patients and those with diabetic ketoacidosis, highlight their role in early detection and risk stratification. Our analysis uniquely demonstrates that these immunological markers are not only strongly associated with T1DM but also correlate with clinical severity, including a markedly higher burden in patients presenting with diabetic ketoacidosis. The strong inverse correlation with age further supports their role in risk stratification across demographic groups. Importantly, the detection of autoantibodies in a subset of T2DM patients, especially those on insulin therapy, reveals likely misclassification and supports the presence of LADA. By linking autoantibody burden with treatment patterns and complication risk, our findings offer novel clinical insights and underscore the importance of integrating autoantibody screening into routine diagnostic workflows. Additionally, the application of neural network modeling further underscores the diagnostic power of combining age, autoantibody data, and metabolic markers. These findings advocate for the routine implementation of targeted autoantibody panels in clinical practice to enhance diagnostic precision, identify LADA, and inform timely, individualized treatment strategies. A notable strength of this study is the use of exploratory modeling techniques, such as TURF and neural networks, to complement standard statistical analysis, which may offer greater insight into cases of diagnostic ambiguity.

Future research should address key limitations, including the cross-sectional design, incomplete antibody data for some patients, and the lack of genetic or environmental risk factor assessment. Longitudinal studies incorporating extended biomarker panels may further enhance the generalizability and predictive strength of these findings.

## Figures and Tables

**Figure 1 diagnostics-15-02296-f001:**
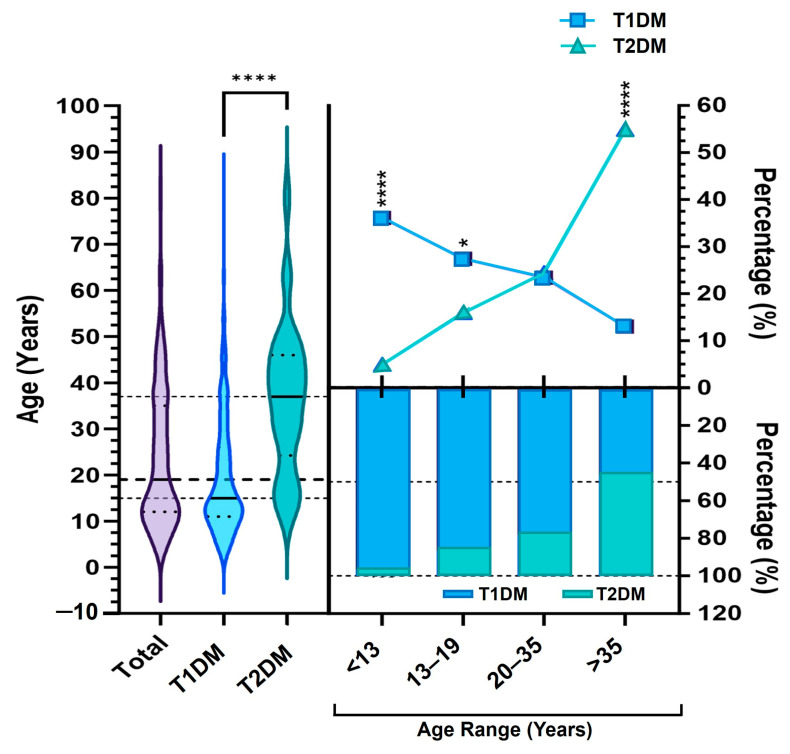
The relationship between age and diabetes subtypes (T1DM and T2DM). The violin plot (left) displays the overall age distribution by diabetes subtype, with horizontal dashed lines marking the predefined age groups: <13, 13–19, 20–35, and >35 years. A significant difference in age distribution between subtypes is observed by the Mann–Whitney test (*p* < 0.0001). The polygon plot (top right) presents the percentage of each diabetes subtype across the four age categories, while the bar plot (bottom right) shows the distribution of patients by age group within each diabetes subtype, and the patients confirmed as T1DM or T2DM (Total). Asterisks denote statistical significance: **** *p* < 0.001; * *p* < 0.05.

**Figure 2 diagnostics-15-02296-f002:**
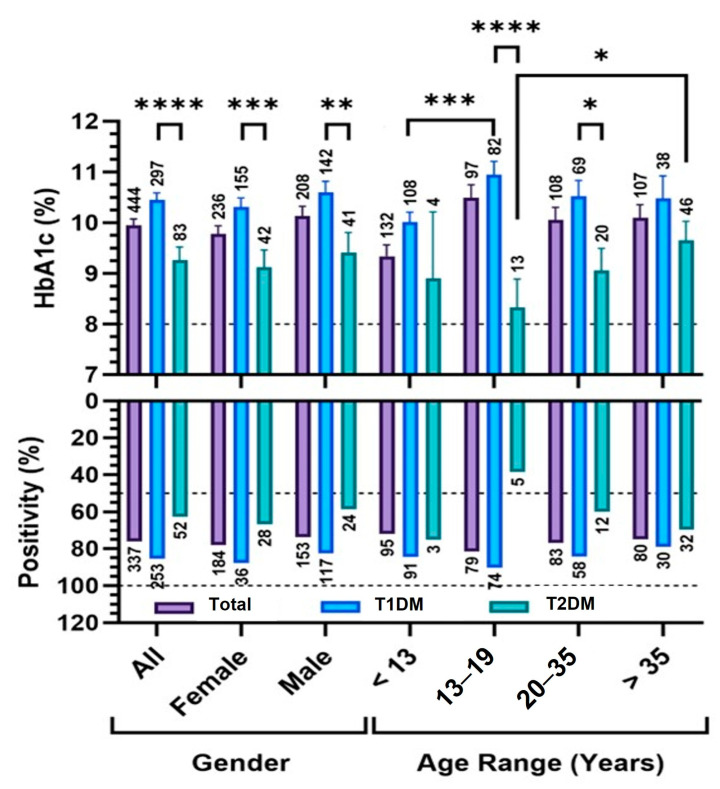
Distribution of HbA1c levels and positivity rates among diabetes patients. The upper panel displays mean HbA1c levels (%), with the dashed line indicating the 8.0% threshold for poor glycemic control. The lower panel shows the percentage of patients exceeding the diagnostic threshold. Data are stratified by sex, age categories (<13, 13–19, 20–35, >35 years), and by diabetes subtype (T1DM and T2DM) (Total). Results are also presented for the broader All group, which includes the entire tested cohort, including unclassified cases (All). Asterisks denote statistical significance: **** *p* < 0.001; *** *p* < 0.005; ** *p* < 0.01; * *p* < 0.05. Error bars represent standard error of the mean (SEM), and sample sizes are indicated above each bar.

**Figure 3 diagnostics-15-02296-f003:**
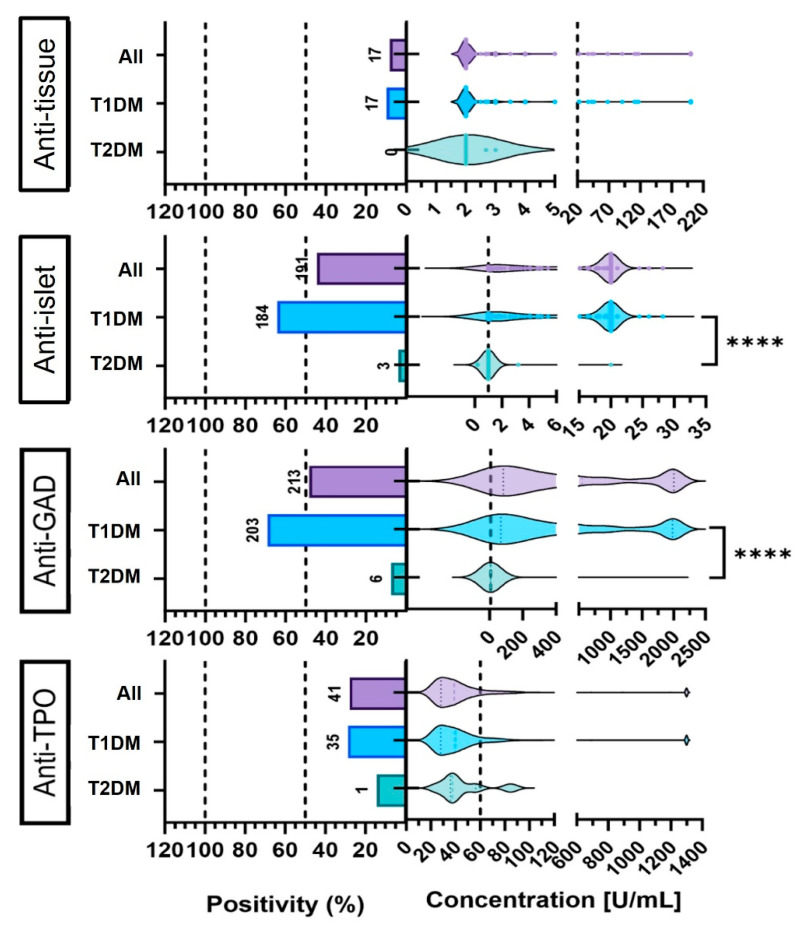
Comparison of autoantibody positivity and levels between T1DM and T2DM patients. Bar plots show the proportion of patients with positive results, based on diagnostic thresholds: anti-GAD (≥5 U/mL), anti-islet (≥1 U/mL), anti-tissue (≥20 U/mL) and anti-TPO (≥60 U/mL). Violin plots illustrate the distribution of measured antibody levels, with dashed lines marking cutoff values and widths indicating data density; inside each violin, the dashed vertical line marks the group median). Small circles overlaid on the violins denote individual patient measurements, slightly offset horizontally to avoid overlap. Box widths reflect group sample sizes. Bar heights represent the frequency of patients above diagnostic thresholds. The “All” group includes all tested patients, including those not definitively classified as T1DM or T2DM. Asterisks denote statistical significance: **** *p* < 0.0001.

**Figure 4 diagnostics-15-02296-f004:**
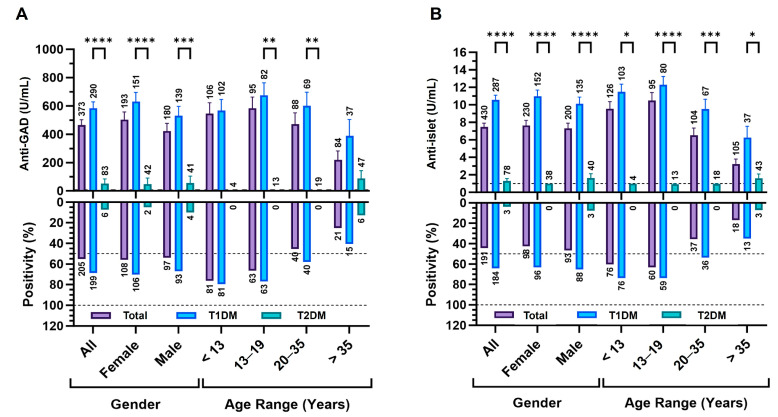
Diagnostic autoantibody profiles in T1DM and T2DM. Anti-GAD (**A**) and anti-islet (**B**) are shown with upper panels displaying mean antibody level (U/mL) with a dashed horizontal marking the manufacturer positivity threshold where shown (anti-GAD ≥ 5 U/mL; anti-islet ≥ 1 U/mL), and lower panels showing positivity (%) with dashed reference lines at 50% and 100%. Data are presented for the entire tested cohort (All; classified and unclassified) and by diabetes subtype (T1DM, T2DM), stratified by sex and age group (<13, 13–19, 20–35, >35 years). Asterisks denote statistical significance: **** *p* < 0.001; *** *p* < 0.005; ** *p* < 0.01; * *p* < 0.05. Error bars represent SEM and sample sizes are shown above each bar.

**Figure 5 diagnostics-15-02296-f005:**
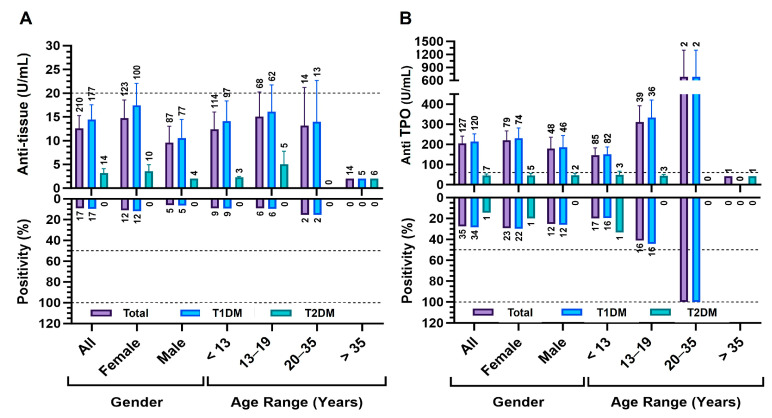
Antibody profiles indicating comorbid autoimmunity in T1DM and T2DM. Anti-tissue (**A**) and anti-TPO (**B**) are shown with upper panels displaying mean antibody level (U/mL) with a dashed horizontal line indicating the manufacturer positivity threshold (anti-tissue ≥ 20 U/mL; anti-TPO ≥ 60 U/mL), and lower panels showing positivity (%) with dashed reference lines at 50% and 100%. Data are presented for the entire tested cohort (All; classified and unclassified) and by diabetes subtype (T1DM, T2DM), stratified by sex and age group (<13, 13–19, 20–35, >35 years). Error bars represent SEM, and sample sizes are shown above each bar.

**Figure 6 diagnostics-15-02296-f006:**
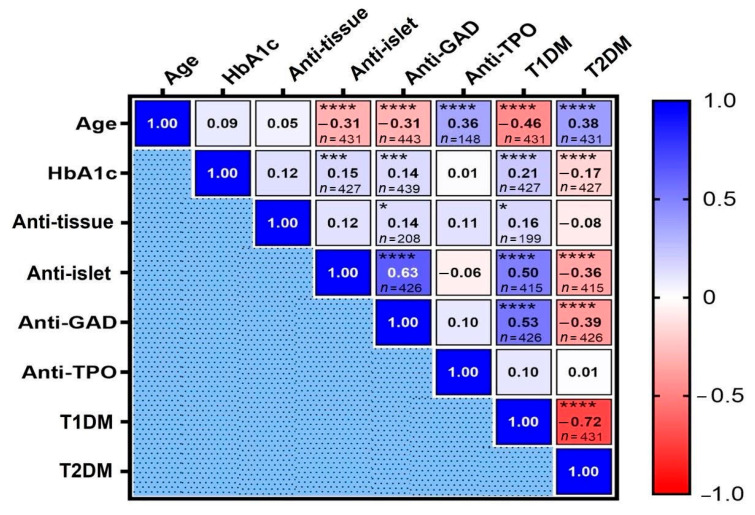
Spearman correlation matrix of clinical and immunological parameters. The heatmap displays Spearman correlation coefficients (r) between clinical variables (age, HbA1c, diabetes subtype) and levels of four diabetes-related autoantibodies (anti-GAD, anti-islet, anti-tissue, and anti-TPO). Diabetes subtypes (T1DM and T2DM) were coded as binary variables (presence = 2, absence = 1) for correlation analysis. Color gradients indicate correlation strength and direction (−1.0 = strong negative, red; +1.0 = strong positive, blue). Asterisks denote significance (* *p* < 0.05; *** *p* < 0.005; **** *p* < 0.001); *N*-values indicate sample sizes for each comparison.

**Figure 7 diagnostics-15-02296-f007:**
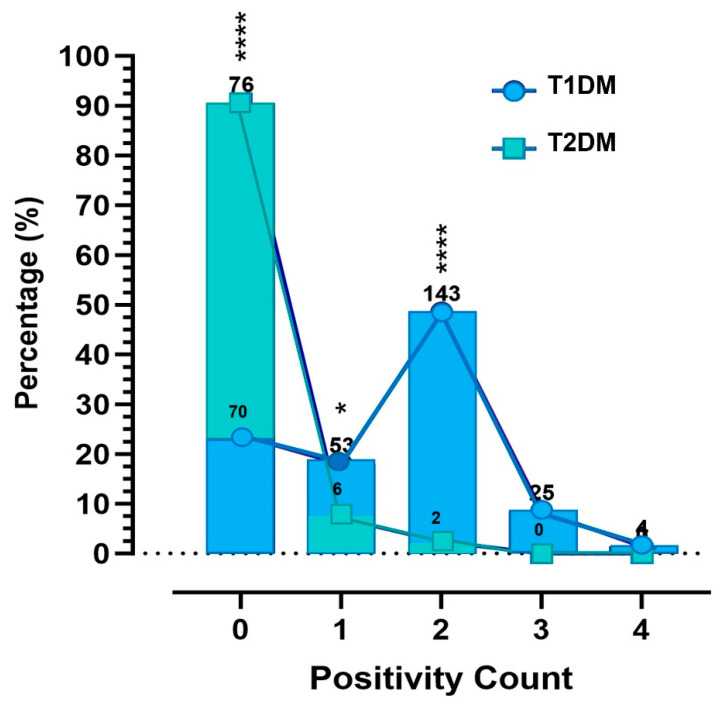
Distribution of autoantibody positivity counts in patients with T1DM and T2DM. Bars represent the percentage of patients testing positive for 0 to 4 autoantibodies. T1DM patients showed a broader range of positivity patterns, while most T2DM patients were antibody-negative. The dashed horizontal line at y = 0% is a reference baseline. Asterisks indicate statistically significant differences between T1DM and T2DM groups (* *p* < 0.05 to **** *p* < 0.0001). Sample sizes are indicated above each bar.

**Figure 8 diagnostics-15-02296-f008:**
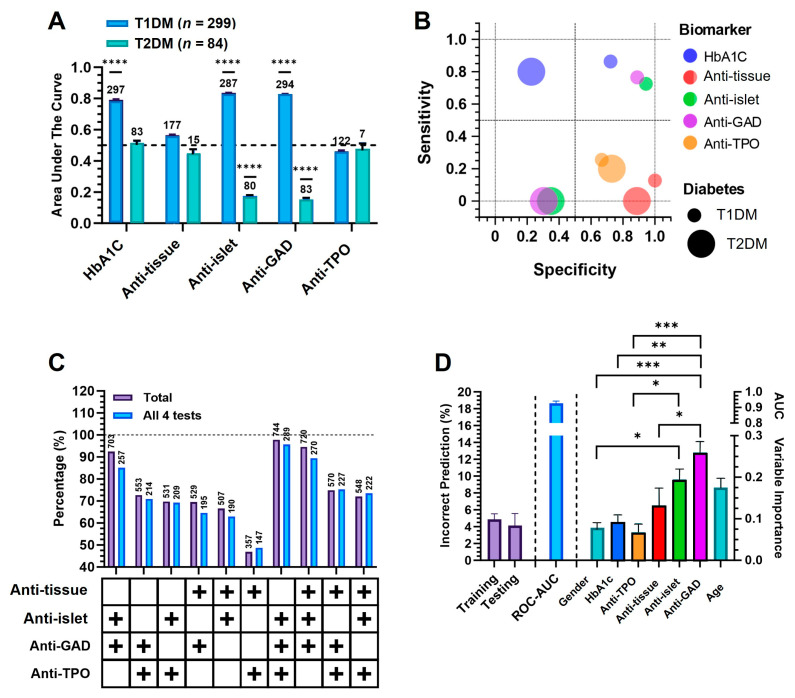
Combined Autoantibody and Predictive Modeling Analysis in T1DM and T2DM. (**A**) Bars show AUC (0–1) for HbA1c, anti-tissue, anti-islet, anti-GAD, and anti-TPO in distinguishing T1DM from T2DM within the classified cohort (299 with T1DM, 84 with T2DM). Numbers above bars denote the per-subtype sample size used to compute the AUC (positives and negatives combined). The dashed line marks AUC = 0.5. (**B**) Bubble plot visualizing diagnostic performance: diagnostic reach (bubble size). Each bubble is color-coded by marker. (**C**) TURF analysis showing the percentage of T1DM patients testing positive for combinations of anti-GAD, anti-Islet, anti-TPO, and anti-tissue antibodies. The “Total” dataset includes all patients tested for at least one marker, while “All 4 tests” includes only those with results available for all four markers. Combinations are indicated by “+” symbols below the x-axis. (**D**) Neural network-based classification of diabetes subtypes: The left panel shows the average error rates during training and testing, with the corresponding AUC plotted on a secondary Y-axis to assess model performance. The right panel presents the mean importance scores of each predictor used in the model. Results are based on five independent runs; error bars represent standard deviations. The dotted horizontal line at 100% indicates the scale maximum and serves as a reference guide. Statistical significance is indicated as * *p* < 0.05, ** *p* < 0.01, *** *p* < 0.005. **** *p* < 0.001.

**Table 1 diagnostics-15-02296-t001:** Analytical Characteristics and Positivity Rates of Diabetes-Related Biomarkers.

Marker	Mean ± SD (*n*) *	Min	Max	Threshold **	Positive% (*n*)	Negative% (*n*)	Notes
HbA1c	10.0 ± 2.6 (444)	4.2	17.6	8.0%	75.9% (337)	24.1% (107)	CV = 19.36% (1604 tests)
Anti-tissue	12.6 ± 38.4 (210)	2.0	200	20 U/mL	8.1% (17)	91.9% (193)	Semi-quantitative ELISA
Anti-islet	7.5 ± 8.6 (430)	0.1	28.2	1 U/mL	44.4% (191)	55.6% (239)	Semi-quantitative ELISA
Anti-GAD	403.7 ± 705.9 (442)	0.5	2000	5 U/mL	48.2% (213)	51.8% (229)	Semi-quantitative ELISA
Anti-TPO	210.5 ± 408.9 (147)	23.5	1300	60 U/mL	27.9% (41)	72.1% (106)	Semi-quantitative ELISA

* *n*. refers to the number of patients who have been tested or have had positive or negative results. ** Threshold values distinguish positive from negative patients for each test.

**Table 2 diagnostics-15-02296-t002:** Clinical, immunological, treatment, and complication characteristics stratified by diabetes subtype.

Marker	T1DM	T2DM	*p* Value
Age (mean ± SD, *N*)	19 ± 13, 299	36 ± 16, 84	<0.001
Sex (%, *n*)	Male: 143 (47.9%) Female: 156 (52.1%)	Male: 41 (48.8%) Female: 43 (51.2%)	–
HbA1c (mean ± SD, *n*)	10.45 ± 2.37, 297	9.26 ± 2.36, 83	<0.001
HbA1c ≥ 8.0% (%, *n*)	253 (83.0%)	52 (17.0%)	<0.001
Mean autoantibody burden (±SD, *n*)	1.47 ± 0.99, 299	0.12 ± 0.39, 84	<0.001
Treatment patterns (%, *n*)	Insulin: 292 (85.1%) Metformin: 37 (42.0%) Sulfonylureas: 5 (71.4%) Sitagliptin: 14 (30.4%)	Insulin: 51 (14.9%) Metformin: 51 (58.0%) Sulfonylureas: 2 (28.6%) Sitagliptin: 32 (69.6%)	<0.001 (Insulin, Metformin, Sitagliptin)
Autoantibody burden by treatment (mean ± SD)	Insulin: 1.74 ± 0.05 Metformin: 0.30 ± 0.05 Sulfonylureas: 0.37 ± 0.13 Sitagliptin: 0.25 ± 0.07	Insulin: 0.53 ± 0.08 Metformin: 0.29 ± 0.06 Sulfonylureas: 0.21 ± 0.08 Sitagliptin: 0.20 ± 0.04	<0.0001 (Insulin); n.s. (others)
Complications prevalence (%, *n*)	DKA: 32.5 (89.4% T1DM) CKD: 3.4 Retinopathy: 4.7 Neuropathy: 2.4 Foot: 1.3	DKA: 10.6% CKD, Retinopathy, Neuropathy, Foot: rare	<0.001 (DKA)
Autoantibody burden by complication (mean ± SD)	DKA: 2.01 ± 0.12 CKD: 0.34 ± 0.06 Retinopathy: 0.50 ± 0.07 Neuropathy: 0.29 ± 0.06 Foot: 0.15 ± 0.05	DKA: 1.00 ± 0.11 Others < 0.6	<0.0001 (DKA)

n.s. refers to results that are not significant (*p* ≥ 0.05).

## Data Availability

This manuscript includes the primary data generated and analyzed during the study. Additional data supporting the findings are available from the corresponding author upon reasonable request and subject to institutional and ethical considerations.

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
