# Peer review of "Autoantibody Profiling for Accurate Differentiation of Type 1 and Type 2 Diabetes Mellitus in Omani Patients: A Retrospective Study"

_diagnostics, 2025, doi:10.3390/diagnostics15182296_

Round 1
Reviewer 1 Report
Comments and Suggestions for Authors
This manuscript presents a retrospective study examining the utility of diabetes-related autoantibodies—anti-GAD, anti-Islet, anti-TPO, and anti-Tissue—in distinguishing T1DM from T2DM in an Omani population. The authors integrate machine learning (neural network modeling) to enhance classification accuracy. The study is clinically relevant, especially in regions where autoimmune diabetes screening is limited. While much of the knowledge is well-established in prior studies, this study contributes regional insight. However, there are several concerns that need to be addressed.
Major comments:
- Beyond the listed eligibility criteria, the manuscript should clarify how patients were selected for inclusion. The study includes 299 T1D and 84 T2D patients—this heavy imbalance raises concerns. Is this distribution consistent with national or regional prevalence data in Oman? The predominance of T1D cases may bias the results and undermine the validity of related analyses or AI model performance.
- The manuscript reports only age, sex, and HbA1c. More comprehensive clinical data are essential for interpreting antibody results and a comparison of these factors between T1D and T2D with a table will clearly show the difference.
- Diabetes duration significantly affects autoantibody levels.
- C-peptide and treatment (insulin or oral agents) are important factors.
- Anti-Islet antibodies are known to be affected by exogenous insulin. How many patients were tested after insulin treatment? Anti-islet in those cases should be either excluded or analyzed separately.
- Types, numbers, and levels of antibodies should be analyzed in relation to diabetes duration.
- Sections 3.10–3.11: The widely accepted diagnostic approach for T1D focuses on islet autoantibodies (GADA, IAA, IA-2A, ZnT8A). Thyroid antibodies may be comorbid but should not be interpreted as biomarker for T1D diagnosis. Both types of antibodies are organ specific.
- Lines 189–191 report strong assay reproducibility (CV < 2% for anti-GAD and anti-Islet). However, the specific assay methods used for each antibody should be clearly described. Since the study is retrospective, explain how CVs were calculated. The use of “concentration” of antibodies is misleading unless a quantitative assay was used. Most immunoassays provide relative signal strength rather than true concentration.
- The inclusion of HbA1c in TURF analysis is questionable. HbA1c is highly variable and influenced by many factors. It is not a reliable biomarker for distinguishing diabetes subtypes.
Minor comments:
- "In the “2.2. Eligibility Criteria”, anti-ICA was included but in “Data collection” it was not mentioned.
- The statement “HbA1c: ≥8 mmol/mol” is unclear. Please convert this to the internationally standardized unit. Additionally, explain how the threshold was determined—was it based on diagnostic criteria, treatment targets, or used solely for stratification?
- The inclusion of the "All" category is inconsistent across figures. In some cases, it equals to the sum of T1D and T2D, in others it’s not. If it did not add more information beyond T1D and T2D, consider removing the category.
- The figures are visually cluttered and hard to read, with excessive minor tick marks and dense formatting.
Author Response
Dear Respected Reviewer,
We sincerely thank you for the thoughtful and detailed feedback on our manuscript entitled “Autoantibody Profiling for Accurate Differentiation of Type 1 and Type 2 Diabetes Mellitus in Omani Patients: A Retrospective Study”. Your constructive comments have greatly contributed to strengthening the scientific rigor and clarity of our work. Below, we provide a point-by-point response. All modifications have been incorporated into the revised manuscript, with additions marked in blue and deletions shown in red for clarity.
Comment 1
Beyond the listed eligibility criteria, the manuscript should clarify how patients were selected for inclusion. The study includes 299 T1D and 84 T2D patients—this heavy imbalance raises concerns. Is this distribution consistent with national or regional prevalence data in Oman? The predominance of T1D cases may bias the results and undermine the validity of related analyses or AI model performance.
Response 1:
We thank the reviewer for this important observation. We clarify that the data for this study were obtained from Sohar Hospital, a tertiary referral hospital, and that the composition of the study cohort reflects the hospital’s referral and testing practices rather than the true population-level prevalence of diabetes mellitus in Oman. In this hospital, most T2DM patients are managed at the primary care level and are referred only for complex or atypical cases requiring further diagnostic evaluation, optimization of management, or initiation of advanced pharmacotherapy. In contrast, most T1DM patients present with diabetic ketoacidosis (DKA), necessitating hospitalization and comprehensive investigation, which increases their likelihood of inclusion. In addition, autoantibody testing is not part of routine screening but is selectively ordered, particularly for pediatric patients or when autoimmune diabetes is suspected, resulting in overrepresentation of T1DM cases in the tested cohort.
To address this, we have incorporated the following revisions into the manuscript:
- Section 2.1 (Study Design and Population, page 3): Added a detailed explanation of Sohar Hospital’s tertiary care role, typical referral patterns for T1DM and T2DM, and the selective ordering of autoantibody testing.
- Discussion (page 25): Added a note highlighting that the predominance of T1DM in our sample is attributable to referral and testing practices at Sohar Hospital, and explaining how this context should inform interpretation of antibody prevalence and the generalizability of the AI model.
Comment 2
The manuscript reports only age, sex, and HbA1c. More comprehensive clinical data are essential for interpreting antibody results and a comparison of these factors between T1D and T2D with a table will clearly show the difference.
- Diabetes duration significantly affects autoantibody levels.
- C-peptide and treatment (insulin or oral agents) are important factors.
- Anti-Islet antibodies are known to be affected by exogenous insulin. How many patients were tested after insulin treatment? Anti-islet in those cases should be either excluded or analyzed separately.
- Types, numbers, and levels of antibodies should be analyzed in relation to diabetes duration.
Response 2:
We appreciate the reviewer’s thoughtful and constructive suggestions. We clarify that this study was designed to evaluate the utility of autoantibody screening in a real-world clinical context, where timely diagnostic insights are critical for guiding treatment decisions. The data collection approach reflects the hospital’s actual clinical workflows, including selective testing practices and case referral patterns. In light of the reviewer’s points, we have provided the following clarifications and corresponding manuscript revisions:
- Diabetes Duration and Autoantibody Levels:
We agree with the reviewer that diabetes duration is an important factor influencing autoantibody persistence and β-cell function. However, in our dataset, onset dates were inconsistently documented, particularly for patients referred from other facilities, making it impossible to accurately incorporate this variable into the analysis without introducing bias.
This limitation has now been explicitly acknowledged in the revised Discussion (limitations paragraph, Page 29), where we also recommend standardized onset-date documentation in future prospective studies to enable robust duration-based analyses.
- C-Peptide and Treatment Data:
While C-peptide is a valuable biomarker for assessing residual β-cell function, in our hospital setting, these tests were not routinely included in the diagnostic workup, particularly for suspected type 1 diabetes mellitus (T1DM). C-peptide testing was requested only in a limited number of patients, most of whom had a documented history of hypoglycemia or presented with diagnostic uncertainty. As the available C-peptide data were too sparse and selective, they were excluded from the statistical analysis. This rationale has been noted as a limitation in the Discussion (Page 29),
- Insulin Use and Antibody Testing: We confirm that, in all cases, autoantibody testing was performed prior to insulin initiation to avoid potential assay interference from exogenous insulin. This methodological detail has been explicitly added to the Materials and Methods, Section 2.4. Data Collection (page 4) as follows: “For suspected T1DM cases, autoantibody testing was generally performed prior to insulin initiation to avoid interference with the anti-Islet assay.”
- Types, Numbers, and Levels of Antibodies in Relation to Duration
We agree with the reviewer that correlating antibody types, counts, and titers with diabetes duration would provide valuable insights into disease stage and phenotype differentiation (e.g., early T1DM vs. long-standing LADA vs. T2DM). Unfortunately, as noted in Point 1, inconsistent documentation of onset dates prevented a robust, bias-free stratification by duration in this retrospective dataset. Performing such analyses with incomplete or unreliable data risked introducing error and reducing validity. For this reason, duration-based antibody comparisons were not performed in the present study.
- Expanded results on autoantibody burden:
We thank the reviewer for this valuable suggestion. In response, we have added a new comparative table (Table 2) and corresponding text in the Results (section 3.12., pages 19–20). This table summarizes key clinical and immunological characteristics of patients with T1DM and T2DM, including age, sex, HbA1c (mean ± SD), autoantibody profiles (anti-GAD, anti-Islet, anti-TPO, anti-Tissue), and treatment modalities. The inclusion of p-values highlights statistically significant differences between the groups. This addition provides a clearer illustration of the phenotypic and immunological distinctions between the two diabetes subtypes, enhancing the overall interpretability of our findings.
Comment 3
Sections 3.10–3.11: The widely accepted diagnostic approach for T1D focuses on islet autoantibodies (GADA, IAA, IA-2A, ZnT8A). Thyroid antibodies may be comorbid but should not be interpreted as biomarker for T1D diagnosis. Both types of antibodies are organ specific.
Response 3
We thank the reviewer for this valuable and accurate observation. We fully agree that the established diagnostic framework for T1DM relies primarily on islet-specific autoantibodies such as GADA, IAA, IA-2A, and ZnT8A. In our study, the inclusion of anti-TPO and anti-tissue antibodies was not intended to imply direct diagnostic utility for T1DM but rather to assess the broader immunological landscape and the coexistence of organ-specific autoimmunity in individuals with diabetes.
To address this point, we have clarified in the Results (Section 3.10, page 15) that these markers did not significantly correlate with islet-specific antibodies and were interpreted as indicators of comorbid autoimmunity rather than diagnostic markers. In the Discussion, at the end of the limitations paragraph (page 29), we have emphasized this distinction and explained that their inclusion was aimed at exploring autoimmune clustering and patient stratification, not at redefining diagnostic criteria.
We believe these clarifications will help prevent misinterpretation of our findings regarding non-islet autoantibodies.
Comment 4
Lines 189–191 report strong assay reproducibility (CV < 2% for anti-GAD and anti-Islet). However, the specific assay methods used for each antibody should be clearly described. Since the study is retrospective, explain how CVs were calculated. The use of “concentration” of antibodies is misleading unless a quantitative assay was used. Most immunoassays provide relative signal strength rather than true concentration.
Response 4
Thank you for highlighting this important point. We agree that the use of “concentration” and the reporting of CVs for antibody assays can be misleading, especially since these were performed using semi-quantitative ELISA kits that report signal intensity rather than absolute concentration. We initially included CV values for all assays to demonstrate assay consistency. However, we acknowledge that this approach was not methodologically appropriate for semi-quantitative immunoassays, which do not provide true concentration values. To address this, the following revisions have been done in the Results section (Section 3.3., page 7 and 8).:
- The original paragraph in the Results section and the corresponding Table 1 have been replaced in full.
- All CV values for antibody assays have been removed; CV is retained only for HbA1c (quantitative method), and the table has been reformatted with explanatory notes instead of CV columns.
- The explanatory footnote “***CV% represents the coefficient of variation for each test; n refers to the number of repetitions used for calculation.” has also been deleted to avoid confusion.
Comment 5
The inclusion of HbA1c in TURF analysis is questionable. HbA1c is highly variable and influenced by many factors. It is not a reliable biomarker for distinguishing diabetes subtypes.
Response 5:
We thank the reviewer for this important observation and fully agree that HbA1c is not a subtype-specific biomarker. Its inclusion in the TURF, ROC, and ANN analyses was exploratory, intended to compare its diagnostic contribution against that of immunological markers and to determine whether it offered any incremental value when combined with autoantibody profiles. To address this, we have:
- Revised the Methods section (Section 2.5, Statistical Analysis, page 5) to clarify that HbA1c was included in the TURF, ROC, and neural network analyses for comparative and exploratory purposes only, with explicit acknowledgment of its limitations in subtype classification.
- Updated the Discussion section (Page 27) to note that, as expected, HbA1c had limited discriminatory power in distinguishing T1DM from T2DM, reinforcing the importance of immunological profiling for accurate classification. However, its inclusion helped contextualize metabolic status across subgroups and supported a holistic understanding of disease presentation.
Minor comments:
Comment 1:
"In the “2.2. Eligibility Criteria”, anti-ICA was included but in “Data collection” it was not mentioned.
Response 1:
We thank the reviewer for this valuable observation. We agree that there was an inconsistency between the eligibility criteria and the actual data analyzed. While anti-ICA was initially considered as a potential marker during the early planning phase, it was not available in the electronic medical records for any of the included patients during the study period. Therefore, it was not included in the final data extraction or analysis.
To address this, we have revised the eligibility criteria in Materials and Methods, Section 2.2. (page 3) to clarify that although anti-ICA was initially listed, it was not ultimately included in the dataset.
Comment 2
The statement “HbA1c: ≥8 mmol/mol” is unclear. Please convert this to the internationally standardized unit. Additionally, explain how the threshold was determined—was it based on diagnostic criteria, treatment targets, or used solely for stratification?
Response 2:
We thank the reviewer for pointing out this important clarification. The HbA1c value was mistakenly reported as “≥8 mmol/mol,” which is not a valid expression in the context of HbA1c reporting. We have now replaced the figure 2 and corrected the value to the internationally standardized unit of ≥8.0% (64 mmol/mol) in both the Results and Methods sections.
We also note that elsewhere in the manuscript, HbA1c values were consistently expressed as percentages in accordance with the NGSP standard, and this correction ensures uniformity throughout the text.
This threshold was selected to reflect suboptimal glycemic control and was used solely for stratification purposes during exploratory analysis to evaluate potential associations between glycemic status and autoantibody profiles. It was not used as a diagnostic criterion nor as a treatment target, but rather as a reference point aligned with commonly accepted clinical cut-offs for poor glycemic control in diabetes management guidelines.
Comment 3
The inclusion of the "All" category is inconsistent across figures. In some cases, it equals to the sum of T1D and T2D, in others it’s not. If it did not add more information beyond T1D and T2D, consider removing the category.
Response 3:
We thank the reviewer for this insightful comment. The inclusion of the “All” category in the figures was intentional and aimed at providing a comprehensive overview of the total study population, which complements the subgroup analyses of T1DM and T2DM. In certain analyses, presenting the “All” category offers valuable context for interpreting overall trends and helps visualize patterns that may not be apparent when examining each subtype in isolation. While it may not always equal the exact sum of T1DM and T2DM due to missing data or specific subgroup inclusion, we have retained it to enhance interpretability at the population level. Detailed explanations have been provided in the figure legends to clarify the scope and meaning of each category, including “All,” and to guide the reader in navigating the visual data accurately.
Comment 4:
The figures are visually cluttered and hard to read, with excessive minor tick marks and dense formatting.
Response 4:
We appreciate the reviewer’s feedback. While we recognize that the figures present detailed information, they were intentionally formatted to retain granularity and allow for side-by-side comparisons across variables. The visual structure was designed to ensure clarity while preserving the integrity of the data, especially in the context of subtype differentiation. Explanations have been provided in the figure legends, which have been carefully crafted to guide interpretation without compromising the analytical depth of the visuals.
We believe the revisions have significantly strengthened our study, and we appreciate the reviewer’s time and expertise.
Sincerely,

Reviewer 2 Report
Comments and Suggestions for Authors
The manuscript titled "Autoantibody Profiling for Accurate Differentiation of Type 1 and Type 2 Diabetes Mellitus in Omani Patients: A Retrospective Study" is carefully read and reviewed. Authors studied the diagnostic utility of diabetes-related autoantibodies in an Omani cohort. They also evaluated their predictive performance using machine learning. Authors evaluated anti-GAD, anti-Islet, anti-TPO, and anti-Tissue antibodies in the study. Therefore, they broadened the autoimmune spectrum and enhanced diagnostic depth, especially the detection of polyautoimmunity. Smartly, authors integrated a neural network model to the study. Hence thay demonstrated that anti-GAD, anti-Islet antibodies, and age were the top predictors, achieving >92% classification accuracy.
Despite these strengths, study has also some issue to revise.
- Authors noted that "Patients were classified as T1DM or T2DM based on clinical diagnosis supported by autoantibody profiles". However, details are needed. The methodology didn't specify how patients were initially classified into T1DM vs. T2DM (clinical criteria? insulin use? BMI? age of onset?), raising concerns about circular reasoning if antibody data informed the original diagnosis. Explain in detail please.
- I think authors didn't stratify the results of the study by diabetes duration. It is critical, especially when interpreting antibody persistence and disease phenotype (i.e., LADA vs. long-standing T2DM).
- The study revealed that accuracy of the AI model was 92%. This is a promising finding but the model was not tested on an external or independent validation cohort. Discuss please.
- Authors tested several antibodies in the study but they didn't include other important markers like IA-2 (Insulinoma-Associated Protein 2) and ZnT8 (Zinc Transporter 8), which could improve sensitivity. Discuss please.
- The absence of C-peptide levels or genetic markers (i.e., HLA typing) weakens the ability to biochemically confirm autoimmune destruction or residual insulin secretion, which is important in differentiating atypical diabetes cases. Discuss please.
Author Response
Dear Respected Reviewer,
We are grateful to the Respected Reviewer for the insightful comments and valuable suggestions that have enhanced the quality of our manuscript. Your remarks regarding diagnostic classification, disease duration, biomarker inclusion, and AI model validation were particularly helpful. Below is our point-by-point response. Corresponding revisions have been made in the manuscript, with additions marked in blue and deletions shown in red for clarity.
Comment 1
Authors noted that "Patients were classified as T1DM or T2DM based on clinical diagnosis supported by autoantibody profiles". However, details are needed. The methodology didn't specify how patients were initially classified into T1DM vs. T2DM (clinical criteria? insulin use? BMI? age of onset?), raising concerns about circular reasoning if antibody data informed the original diagnosis. Explain in detail please.
Response 1:
We appreciate the reviewer’s thoughtful comment and agree that a clear explanation of the classification process is essential. o address this, we have revised Section 2.3 (Materials and Methods, page 3) to clarify the basis for diabetes subtype classification and to ensure transparency regarding the use of autoantibody data.
Patients in this study were managed at a tertiary referral center for diabetes over a five-year period. The diabetes subtype (T1DM or T2DM) was clinically assigned by the treating physician at the initial consultation using ICD coding in the electronic medical record system, based on standard diagnostic criteria. This diagnosis was routinely reviewed during follow-up visits, and any confirmed change was documented in the medical notes. We extracted the final diagnosis as recorded unless such changes were explicitly noted.
In our clinical setting, T1DM was typically diagnosed in younger individuals with DKA, ketosis, or rapid symptom onset requiring early insulin therapy, prompting more extensive diagnostic work-up. T2DM was diagnosed in patients with adult-onset hyperglycemia, obesity, and features of metabolic syndrome, who initially responded to oral agents. Notably, obese patients with metabolic syndrome were not routinely investigated for autoimmune diabetes unless therapy failure or clinical red flags arose.
Autoantibody results were analyzed only after classification, serving as independent variables to evaluate their correlation with the clinical diagnosis. This strategy avoided circular reasoning and preserved the integrity of the comparative analysis.
Comment 2:
I think authors didn't stratify the results of the study by diabetes duration. It is critical, especially when interpreting antibody persistence and disease phenotype (i.e., LADA vs. long-standing T2DM).
We appreciate the reviewer’s valuable observation. We fully agree that diabetes duration is an important factor when evaluating autoantibody persistence and distinguishing between subtypes such as latent autoimmune diabetes in adults (LADA) and long-standing T2DM.
Due to the retrospective nature of our study, consistent documentation of diabetes duration was lacking, particularly among patients referred from other healthcare facilities. In many cases, only estimated or incomplete onset dates were available. Including this variable under such conditions would have introduced potential bias and limited the reliability of any duration-based subgroup analysis.
To ensure methodological consistency, we therefore excluded diabetes duration from the current analysis. This limitation has now been acknowledged in the revised Discussion (limitations paragraph, Page 28), along with a recommendation for future prospective studies to incorporate standardized documentation of disease onset to better assess duration-related immunological trends.
Comment 3
The study revealed that accuracy of the AI model was 92%. This is a promising finding but the model was not tested on an external or independent validation cohort. Discuss please.
Response 3:
We thank the reviewer for this insightful comment. We fully agree that external validation is essential to assess the generalizability and robustness of predictive models, especially those developed using AI methodologies.
In this study, we employed internal validation by randomly splitting the dataset into training (70%) and testing (30%) sets, repeated across five iterations to evaluate model stability. This yielded consistent performance, with a mean AUC of 0.927 and a testing error rate of 4.6%. However, we recognize that internal validation cannot substitute for testing on an independent dataset.
This limitation has now been acknowledged in the revised Discussion section, page 28. We have noted that although the model demonstrated strong predictive capability within our dataset, external validation in larger and more diverse populations is necessary before clinical application. Future studies should focus on multi-center validation to ensure broader applicability and performance consistency.
Comment 4
Authors tested several antibodies in the study but they didn't include other important markers like IA-2 (Insulinoma-Associated Protein 2) and ZnT8 (Zinc Transporter 8), which could improve sensitivity. Discuss please.
Response 4:
We thank the reviewer for this important observation. We fully agree that IA-2 and ZnT8 are established islet autoantibodies that can significantly enhance the sensitivity and diagnostic yield when identifying autoimmune diabetes, especially in cases where anti-GAD or anti-Islet antibodies may be negative.
In the present study, however, our analysis was limited by the retrospective nature of data collection and the clinical testing practices at the study site. At Sohar Hospital, autoantibody testing is not routinely performed for all patients with diabetes but is typically requested on a case-by-case basis when autoimmune diabetes is suspected. During the study period, IA-2 and ZnT8 were not ordered by clinicians in either the adult or pediatric departments, and as such, no data were available for inclusion in the analysis.
We have now acknowledged this in the revised Discussion (section, page 27) as a limitation of the study. We have also highlighted that future prospective investigations incorporating a broader panel of islet autoantibodies, including IA-2 and ZnT8, would be valuable for improving diagnostic sensitivity and better characterizing autoimmune diabetes phenotypes.
Comment 4
The absence of C-peptide levels or genetic markers (i.e., HLA typing) weakens the ability to biochemically confirm autoimmune destruction or residual insulin secretion, which is important in differentiating atypical diabetes cases. Discuss please.
Response 4:
We thank the reviewer for highlighting this important point. We agree that both C-peptide and HLA typing are valuable tools for confirming autoimmune β-cell destruction and assessing residual insulin secretion, particularly in atypical or overlapping diabetes phenotypes.
However, in our clinical setting at Sohar Hospital, these investigations are not part of the standard diagnostic workflow. C-peptide testing is typically reserved for cases with persistent hypoglycemia or diagnostic uncertainty, and was therefore available in only a limited number of patients. HLA typing, while informative in research settings, is not routinely performed in clinical practice and was not documented in the electronic medical records used for this retrospective study.
Our study was designed to evaluate the utility of diabetes-associated autoantibodies in a real-world clinical context, where clinicians often make subtype determinations based on clinical presentation and selectively ordered tests. To reflect this practice accurately, we focused on the subset of patients in whom autoantibody testing was performed during routine care. Where available, antibody testing was conducted prior to insulin initiation to avoid confounding effects of treatment on serological markers.
We have now acknowledged this limitation in the revised Discussion (limitations paragraph, page 28), noting that the absence of C-peptide and HLA data restricts our ability to confirm autoimmune pathophysiology or assess residual insulin production. We also emphasize that future prospective studies incorporating these biomarkers alongside autoantibody panels would enhance the characterization of diabetes subtypes, particularly for identifying latent autoimmune diabetes in adults (LADA) or monogenic forms.
We express our gratitude to Reviewer 2 for the thoughtful suggestions and encouraging remarks. The feedback was instrumental in refining our analysis and presentation. We hope the revised version now addresses all concerns satisfactorily.
Round 2
Reviewer 1 Report
Comments and Suggestions for Authors
The authors have provided detailed responses to the comments and the revisions made to the manuscript were clearly marked. There are still several areas in the revised version that require correction or further refinement.
- Data collection section (Lines 130–146) –Please clarify the collected data of age at what point. At diabetes diagnosis, at hospital admission, or at antibody testing?
- In Methods (Lines 126–128), patients are stratified into four age groups (2–12, 13–19, 20–50, ≥51), but in Results the age categories are different, “<13, 13–19, 20–35, >35”, “The ages of patients ranged from 1 to 83 years” . Please unify the age grouping strategy.
- Clarify why the HbA1c 8.0% cut-off was chosen. There are also typographical errors at Line 258 “90.2.8%”.
- Line 242-245, “Based on positivity patterns, the tests were broadly categorized into three groups:(1) HbA1c with consistently high positivity; (2) anti-GAD and anti-Islet antibodies with balanced positive/negative rates; and (3) anti-Tissue and anti-TPO antibodies with low overall positivity (Table 1).” I didn’t get what’s the meaning of these categories is and there was no further analysis for these three groups.
- The term “concentration” should be avoided for semi-quantitative assays.
- The authors introduced the ANN model “The dataset was randomly partitioned into training (70%) and testing (30%) subsets” but didn’t provide the details. How the neural network was trained and validated, and what software was used?
- A lot of duplicated text, such as sentences in Lines 387–390, 391–396, and 397–402, 494–496, and more. Please check the entire text thoroughly.
- Most figures remain visually busy. Converting some complex figures into tables may improve clarity and interpretability. Too many comparison bars and unclear which comparison bars correspond to which group comparisons. For antibody level plots, please show assay cut-off values and explain it in figure legends. Figure legends should explicitly define all symbols and comparisons, also explain what the lines in the violin plots represent. In Figure 3, anti-Tissue in T2D is shown as 0%, yet the violin plot seemed not, please clarify. Figure 14A: ROC curves may be a more appropriate and interpretable presentation.
Author Response
Dear Respected Prof.,
We would like to thank the reviewer for the insightful and constructive comments, which have greatly contributed to improving the quality and clarity of our manuscript. Below, we provide point-by-point responses to each comment. Revisions made in the manuscript are clearly indicated with corresponding line numbers and section references where applicable. We have carefully addressed all concerns and revised the manuscript accordingly.
Comment 1:
Data collection section (Lines 130–146) – Please clarify the collected data of age at what point. At diabetes diagnosis, at hospital admission, or at antibody testing?
Response 1:
We appreciate the reviewer’s valuable comment. The timing of age documentation has now been clarified in the Data Collection subsection of the Methods section (lines 132, 135, 136, page 4). The revised text specifies that age was recorded at the time of autoantibody testing, which in most cases occurred shortly after diagnosis and before insulin initiation. We hope this addresses the concern.
Comment 2:
In Methods (Lines 126–128), patients are stratified into four age groups (2–12, 13–19, 20–50, ≥51), but in Results the age categories are different, “<13, 13–19, 20–35, >35”, “The ages of patients ranged from 1 to 83 years”. Please unify the age grouping strategy.
Response 2:
We thank the reviewer for highlighting this important inconsistency. To ensure clarity and alignment across the manuscript, we have revised the age groupings in the Classification and Stratification subsection of the Methods section (lines 128, 179, page 4) to reflect the same categories used consistently in the Results and Discussion: <13, 13–19, 20–35, and >35 years. We have also corrected the minimum age in the Results section from “1–83 years” to “2–83 years,” as the youngest patient included in the dataset was 2 years old (lines 224, page 6).
Comment 3:
Clarify why the HbA1c 8.0% cut-off was chosen. There are also typographical errors at Line 258 “90.2.8%”.
Response 3:
We thank the reviewer for this observation. The HbA1c threshold of ≥8.0% was selected as it is frequently used in clinical practice to indicate suboptimal glycemic control and higher risk for diabetes-related complications. While ≥6.5% is the diagnostic cut-off for diabetes, the ≥8.0% level is often applied to stratify patients with poor control who may benefit from treatment adjustment. This clarification has now been included in the Data Collection subsection of the Methods section (lines 139-141, page 4).
Concerning the typographical error, we agree with the reviewer. The number “90.2.8%” was an unintended formatting issue. It has been corrected to “90%” in the revised manuscript for clarity and consistency with the surrounding narrative (line 283, page 8).
Comment 4:
Line 242–245: “Based on positivity patterns, the tests were broadly categorized into three groups: (1) HbA1c with consistently high positivity; (2) anti-GAD and anti-Islet antibodies with balanced positive/negative rates; and (3) anti-Tissue and anti-TPO antibodies with low overall positivity (Table 1).” I didn’t get what’s the meaning of these categories and there was no further analysis for these three groups.
Response 4:
We acknowledge the reviewer’s valuable observation. The classification was intended purely for descriptive clarity, to highlight general trends in marker positivity rates across the cohort. These groupings were not part of an analytical framework or hypothesis testing. To address this point, we have revised the relevant sentence in the Results section (lines 269-270, Page 8) to clarify this purpose explicitly.
Comment 5:
The term “concentration” should be avoided for semi-quantitative assays.
Response 5:
Thank you for your valuable comment. We acknowledge that the term “concentration” is not appropriate when describing results obtained from semi-quantitative assays. In response, we have carefully revised the manuscript and replaced all instances of “concentration” with the term “level”, which more accurately reflects the nature of the data. This revision has been applied consistently across the main text, figure labels, and legends to ensure clarity and accuracy in terminology.
Comment 6:
The authors introduced the ANN model “The dataset was randomly partitioned into training (70%) and testing (30%) subsets” but didn’t provide the details. How the neural network was trained and validated, and what software was used?
Response 6:
We thank the reviewer for this important observation. In response, we have now elaborated on the ANN model in the Methods section (Page 6, Lines 199–206), including the structure of the network, the training algorithm employed, the internal validation strategy used to prevent overfitting, and the software environment in which the model was developed. These clarifications aim to enhance the transparency and reproducibility of the methodology. These additions address the training, validation, and software environment as requested and reflect standard clinical ANN modeling practices.
Comment 7:
A lot of duplicated text, such as sentences in Lines 387–390, 391–396, and 397–402, 494–496, and more. Please check the entire text thoroughly.
Response 7:
Thank you for highlighting this important point. We have conducted a comprehensive review of the manuscript and addressed the instances of unnecessary repetition, specifically in the sections noted (Lines 387–390, 391–396, 397–402, and 494–496). Redundant or overlapping content has been revised or removed to enhance clarity, streamline the narrative, and avoid duplication.
In addition, to better reflect the scope and emphasis of the study, the title has been revised to:
“Autoantibody Profiles and Predictive Modeling in Diabetes Subtypes: A Retrospective Study in Oman”. This revised title more accurately conveys the dual focus on immunological profiling and machine learning analysis, as well as the retrospective and regional context of the research.
Furthermore, the entire manuscript has been revised to eliminate redundancy, ensure consistent use and explanation of all abbreviations and symbols, and improve the overall coherence and readability in line with reviewer feedback.
Comment 8
We thank the reviewer for the thoughtful and constructive feedback on the clarity and presentation of the figures. In response, we have carefully reviewed and revised the relevant sections and figure legends to enhance readability, interpretability, and consistency throughout the manuscript. The following changes and clarifications were made:
Response 8:
We sincerely thank the reviewer for their constructive comments regarding the visual presentation and clarity of our figures. We recognize that some figures appeared visually complex due to multiple subgroup comparisons and overlapping graphical elements. To address these concerns and enhance interpretability, we carefully revised several figures, streamlined the number of comparisons, clarified legends, and reorganized certain elements into tables or supplementary material. Below, we detail the specific actions taken in response to each of the reviewer’s suggestions.
- Simplification of Visuals
- Figures 2, 4, and 5 were redesigned by removing comparisons involving the total cohort, retaining only the contrasts between diabetes subtypes (T1DM vs. T2DM), which are the primary focus of the study.
- Figures 13, 14, and 15 have been consolidated into a single integrated figure, now presented as Figure 14, titled “Combined Autoantibody and Predictive Modeling Analysis in T1DM and T2DM. The corresponding results section has been updated and retitled as 13. “Integrated Autoantibody and Predictive Modeling Analysis” (lines 662–690, pages 27 and 28) to reflect this consolidation. This integration enhances the narrative coherence, minimizes visual redundancy, and maintains all critical scientific content.
- Conversion to Tables and Reorganization
- Figures 11 and 12 were removed from the main manuscript to streamline the presentation and reduce visual complexity, as suggested. Both figures have been moved to the Supplementary Material and are now labeled Supplementary Figures S2 and S3, respectively. The data they contain remain essential and accessible to readers interested in treatment- and complication-specific trends. In parallel, their core content has been consolidated into an integrated summary within the revised Table 2 in the Results section (Section 3.12: Autoantibody Burden Across Treatment Modalities and Complications, lines 581–605, page 24 and 24), which provides a concise yet comprehensive overview of the clinical, immunological, and treatment-related variables stratified by diabetes subtype.
- Individual autoantibody data (e.g., anti-GAD, anti-TPO, anti-Tissue, anti-Islet) were moved to a new Supplementary Table (Supplementary Table S1) to avoid redundancy with visual figures.
- Figure 10 has been moved to the Supplementary Material (now Supplementary Figure S1). This figure illustrates the mean number of positive autoantibodies stratified by sex and age groups in T1DM and T2DM, complementing the subgroup analysis in the main text. The relevant reference to this figure is now included in the Results section titled “3.11. Patterns and Burden of Autoantibody Positivity by Diabetes Subtype” (page 18, lines 488–491).
- Clarification of Group Comparisons and Symbols
- To address the issue of excessive comparison bars, we reduced the number of displayed contrasts to highlight only the most clinically relevant comparisons.
- Figure legends were revised throughout to:
- Clearly define all symbols and colors.
- Indicate which groups are being compared.
- Explain features such as cut-off lines and components of violin plots.
- Clarification of Discrepancy in Figure 3
In Figure 3, the bar plot for anti-Tissue in T2DM shows 0% because none of the patients met the positivity threshold. However, the violin plot displays the distribution of all signal values, including sub-threshold levels. This has been clarified in both the Results section (Page 11, Lines 314–316) and the figure legend.
- Justification for Bar Plot in Figure 14A
While ROC curves are widely used for evaluating model performance, Figure 14A was intended to illustrate subgroup differences in anti-GAD levels rather than assess predictive accuracy. To reflect this, we retained the bar/error plot format and revised the figure legend and Results text (Page 12, Lines 345–346) to clarify its descriptive purpose.
Reviewer 2 Report
Comments and Suggestions for Authors
Authors adequately revised their manuscript. The response of the authors to the issues raised in the previous review round was satisfactory. Well done.
Author Response
We sincerely thank Reviewer 2 for the kind and encouraging feedback. We truly appreciate your thoughtful review and are pleased that the revisions and clarifications were satisfactory. Your positive remarks are highly motivating and helped us improve the clarity and impact of the manuscript.
Round 3
Reviewer 1 Report
Comments and Suggestions for Authors
The authors have made a thorough revision of the manuscript and have addressed most of the previous comments. The revised version shows substantial improvement and overall reads much better. However, several issues remain that need clarification or further refinement:
- Line 269–270: The revision is reasonable; however, consider deleting these classifications as they were not part of an analytical framework or hypothesis testing.
- ANN model: The revision of the ANN model looks good, please clarify whether the model was trained and tested using the same study cohort, and the number of cases included in training and testing.
- Figure 1: The figure remains overly busy and crowded. Consider keeping the same color scheme for T1D and T2D across all three panels and omitting redundant legends. Some information such as sample size numbers (right lower figure) could be moved to the figure legend to simplify the figure. Why the age group 13–19 years includes both categories I and II.
- Figures 4–7: These figures and their corresponding results could be combined, as they all present antibody data with similar descriptions.
- Line 102: “anti-IAA” should be “anti-ICA.”
- Line 252: Please unify the units of HbA1c throughout the manuscript, not sometimes in mmol/mol and sometimes in %.
- Unify naming “anti-tissue” antibodies throughout the manuscript.
- Line 253: Kolmogorov–Smirnov tests should be mentioned in the statistics section.
- Line 314: The sentence “Although no T2DM patients were positive for anti-tissue antibodies, sub-threshold levels were detected in a few individuals, as reflected in the violin plot distribution” is unnecessary and can be removed.
- Line 327-330: Repeated description.
- Line 596: If all clinical data were collected at the time of diabetes diagnosis, it is expected that the prevalence of chronic complications would be low.
- Figure 14A: The legend states that “The number of positive cases per diabetes subtype is displayed above each bar, with total positive cases per test listed next to the test name.” However, there appear to be discrepancies, for example, HbA1c shows a total of 337 positives, yet 297 positives in T1D and 84 positives in T2D (297+84=381). Similar for other tests. Please check and explain these discrepancies.
Author Response
Dear Respected Reviewer,
We thank you for the thoughtful and constructive feedback, which substantially improved the clarity and rigor of the manuscript. Below we respond to each comment in order, indicating the exact section/subsection, page, and line numbers in the revised (with figures renumbered where applicable). Revisions focus on simplifying figures and legends, standardizing terminology/units, clarifying statistical methods and the ANN workflow, and refining text for consistency, without altering the study’s conclusions.
Point-by-Point Responses:
Comment 1
“Line 269–270: The revision is reasonable; however, consider deleting these classifications as they were not part of an analytical framework or hypothesis testing”.
Response 1
We thank the reviewer for this helpful suggestion. The post-hoc three-category classification and its corresponding explanatory sentence have been removed from Section 3.3 (page 9, lines 292–293) of the Results. The Results now report each biomarker directly using the manufacturer’s positivity thresholds, without ad hoc grouping or additional inferential claims.
Comment 2
“The revision of the ANN model looks good; please clarify whether the model was trained and tested using the same study cohort, and the number of cases included in training and testing”.
Response 2
Thank you for the helpful request. We have clarified these details in Section 2.5 of the Methods (page 6, lines 197–204). Specifically, the study cohort (n = 448) was randomly split at the patient level using stratified sampling by diabetes subtype into non-overlapping training (70%; n = 314) and testing (30%; n = 134) sets. All predictors were standardized using parameters estimated on the training set and applied unchanged to the test set. Model selection and tuning were performed exclusively on the training data, and the held-out test set was evaluated once to report prediction error rate, cross-entropy loss, and AUC.
Comment 3
“Figure 1 is overly busy/crowded. Keep the same color scheme for T1DM/T2DM across all panels and omit redundant legends. Some information, such as sample‐size numbers (right lower figure), could be moved to the figure legend to simplify the figure. Why does the age group 13–19 years include both categories I and II?”
Response 3
Thank you for the suggestions. We revised Figure 1, its legend, and the accompanying text to improve clarity and reduce visual load:
- Unified styling: One consistent T1DM/T2DM color scheme across panels; redundant, panel-level legends removed in favor of a single legend
- Decluttering: Removed on-figure sample-size labels and “I/II” markers; axes show only predefined age groups.
- Placement of counts: Sample sizes are now reported in the Results text (Section 3.2, page 7, lines 251-256), not inside the graphic, to keep the figure clean.
- Figure 1 legend: Simplified to a neutral, panel-only description per the reviewer’s request, removed significance statements and “I/II” markers, omitted on-figure/sample-size notes (n now reported in Results subsection 3.2), and clarified that “Total” denotes the combined, confirmed T1DM + T2DM cohort.
Comment 4
“Figures 4–7: These figures and their corresponding results could be combined, as they all present antibody data with similar descriptions”.
Response 4
Thank you for the suggestion. Because the antibody panels address two distinct aims, anti-GAD/anti-islet for diagnostic profiling and anti-tissue/anti-TPO for comorbid autoimmunity, we consolidated by purpose rather than into a single omnibus figure.
Revisions implemented:
- Figure 4 (new), Diagnostic autoantibody profiles in T1DM and T2DM: prior Figures 4 (anti-GAD) and 5 (anti-islet) are combined as panels A–B; the corresponding text appears in the Results, subsection 3.6, “Anti-GAD and anti-islet diagnostic profiles across age and sex” (page 16, lines 413–426). The figure title and legend were updated accordingly.
- Figure 5 (new), Antibody profiles indicating comorbid autoimmunity in T1DM and T2DM: prior Figures 6 (anti-tissue) and 7 (anti-TPO) are combined as panels A–B; the corresponding text appears in the Results, subsection 3.7, “Antibody profiles indicating comorbid autoimmunity in T1DM and T2DM” (page 18, lines 480–496). The figure title and legend were updated accordingly.
To prevent redundancy, the accompanying Results text was condensed to two concise, comparative paragraphs aligned exactly with Figure 4 (subsection 3.6) and Figure 5 (subsection 3.7). This preserves interpretability while improving readability.
Comment 5
“Line 102: “anti-IAA” should be “anti-ICA”.
Response 5:
Thank you for the correction. The prior reference to anti-IAA was misleading in this context. We have removed that note from Materials and Methods, subsection 2.2 (page 3, lines 103–105) and revised the inclusion criteria to list only the antibodies actually analyzed in this study: anti-GAD (GADA), anti-islet (islet cell antibody; ICA), anti-tissue transglutaminase IgA (anti-tTG IgA; hereafter “anti-tissue”), and anti-TPO.
Comment 6:
“Please unify the units of HbA1c throughout the manuscript, not sometimes in mmol/mol and sometimes in %.”
Response 6:
Thank you for highlighting this. We made the specific correction in Results subsection 3.3 (page 8, line 274): “HbA1c: 9.95 ± 2.6 mmol/mol” has been corrected to “HbA1c: 9.95 ± 2.6%”. We then reviewed the entire manuscript to ensure HbA1c is reported uniformly in percent (%) throughout.
Comment 7
“Unify naming ‘anti-tissue’ antibodies throughout the manuscript.”
Response 7
We appreciate this comment. Terminology has been standardized throughout. At first mention, we now use IgA anti–tissue transglutaminase (anti-tTG IgA); thereafter, anti-tissue (lowercase) is used consistently. All legacy variants (“anti-Tissue”) have been corrected. Figure captions and in-figure text (panel labels, axes, keys), including the Figure 8 legend (page 30, line 778) have been updated accordingly.
Comment 8
Line 253: Kolmogorov–Smirnov tests should be mentioned in the statistics section.
Response 8
Thank you for the helpful suggestion. We have added this clarification to Materials and Methods (subsection 2.5, page 5, lines 159–161), noting that normality of continuous variables was assessed using the Kolmogorov–Smirnov test and that these diagnostics guided the choice of parametric vs. nonparametric procedures. This clarification aligns with the corresponding statement in the Results that reports the K–S findings.
Comment 9
“Although no T2DM patients were positive for anti-tissue antibodies, sub-threshold levels were detected in a few individuals, as reflected in the violin plot distribution” is unnecessary and can be removed.”
Response 9
Agreed. The sentence has been deleted to avoid redundancy.
Comment 10
“Line 327-330: Repeated description”.
Response 10
We appreciate the reviewer’s observation. The Figure 3 caption has been streamlined by removing the duplicated sentence (“Violin plots illustrate the distribution of measured antibody levels, with dashed lines marking cutoff values.”). This change is reflected at lines 351–352.
Comment 11
“Line 596: If all clinical data were collected at the time of diabetes diagnosis, it is expected that the prevalence of chronic complications would be low.”
Response 11
Thank you for this helpful observation. We have clarified in Methods (subsection 2.4, page 4, lines 137–142) that the study is a retrospective review of EMR encounters (2020–2024) rather than a uniform baseline-at-diagnosis assessment. Accordingly, chronic complications were recorded when documented within the study window (no standardized baseline screening), whereas DKA reflects acute presentation at hospital entry. This study design explains the low observed prevalence of chronic complications.
Comment 12
Figure 14A: The legend states that “The number of positive cases per diabetes subtype is displayed above each bar, with total positive cases per test listed next to the test name.” However, there appear to be discrepancies; for example, HbA1c shows a total of 337 positives, yet 297 positives in T1D and 84 positives in T2D (297+84=381). Similar for other tests. Please check and explain these discrepancies.
Response 12
Thank you for flagging the apparent discrepancies. The original panel mixed two different quantities and cohorts: bar-top numbers reflected the analysis sample size per subtype used for AUC in the classified cohort (Total), whereas the parenthetical “n” on the x-axis referred to positive counts from the broader All cohort. This led to mismatched totals (e.g., 297 + 84 vs 337). We have resolved this by removing the “n” values from the x-axis and clarified the legend. Panel A now (lines 771-775, legend figure 8 after updating the number of figures) states that numbers above bars are the assay-specific per-subtype sample sizes used to compute AUC in the classified cohort (Total; n = 383; 299 with T1DM, 84 with T2DM), not counts of positives. This eliminates mixing of quantities from different cohorts/definitions.